# Community Exploration: From Offline Optimization to Online Learning

**Xiaowei Chen[1], Weiran Huang[2], Wei Chen[3], John C.S. Lui[1]**
[1]The Chinese University of Hong Kong
[2]Huawei Noah's Ark Lab, [3]Microsoft Research
[1]{xwchen, cslui}@cse.cuhk.edu.hk, [2]huang.inbox@outlook.com
[3]weic@microsoft.com

## Abstract

We introduce the community exploration problem that has many real-world applications such as online advertising. In the problem, an explorer allocates limited budget to explore communities so as to maximize the number of members he could meet. We provide a systematic study of the community exploration problem, from offline optimization to online learning. For the offline setting where the sizes of communities are known, we prove that the greedy methods for both of non-adaptive exploration and adaptive exploration are optimal. For the online setting where the sizes of communities are not known and need to be learned from the multi-round explorations, we propose an "upper confidence" like algorithm that achieves the logarithmic regret bounds. By combining the feedback from different rounds, we can achieve a constant regret bound.

## 1 Introduction

In this paper, we introduce the community exploration problem, which is abstracted from many real-world applications. Consider the following hypothetical scenario. Suppose that John just entered the university as a freshman. He wants to explore different student communities or study groups at the university to meet as many new friends as possible. But he only has a limited time to spend on exploring different communities, so his problem is how to allocate his time and energy to explore different student communities to maximize the number of people he would meet.

The above hypothetical community exploration scenario can also find similar counterparts in serious business and social applications. One example is online advertising. In this application, an advertiser wants to promote his products via placing advertisements on different online websites. The website would show the advertisements on webpages, and visitors to the websites may click on the advertisements when they view these webpages. The advertiser wants to reach as many unique customers as possible, but he only has a limited budget to spend. Moreover, website visitors come randomly, so it is not guaranteed that all visitors to the same website are unique customers. So the advertiser needs to decide how to spend the budget on each website to reach his customers. Of course, intuitively he should spend more budget on larger communities, but how much? And what if he does not know the user size of every website? In this case, each website is a community, consisting of all visitors to this website, and the problem can be modeled as a community exploration problem. Another example could be a social worker who wants to reach a large number of people from different communities to do social studies or improve the social welfare for a large population, while he also needs to face the budget constraint and uncertainty about the community.

In this paper, we abstract the common features of these applications and define the following community exploration problem that reflects the common core of the problem. We model the problem with $m$ disjoint communities $C_1, \ldots, C_m$ with $C = \cup_{i=1}^m C_i$, where each community $C_i$ has $d_i$

members. Each time when one explores (or visit) a community $C_i$, he would meet one member of $C_i$ uniformly at random.[1] Given a budget $K$, the goal of community exploration is to determine the budget allocation $\boldsymbol{k} = (k_1, \ldots, k_m) \in \mathbb{Z}_+^m$ with $\sum_{i=1}^{m} k_i \leq K$, such that the total number of distinct members met is maximized when each community $C_i$ is explored $k_i$ times.

We provide a systematic study of the above community exploration problem, from offline optimization to online learning. First, we consider the offline setting where the community sizes are known. In this setting, we further study two problem variants — the non-adaptive version and the adaptive version. The non-adaptive version requires that the complete budget allocation $\boldsymbol{k}$ is decided before the exploration is started, while the adaptive version allows the algorithm to use the feedback from the exploration results of the previous steps to determine the exploration target of the next step. In both cases, we prove that the greedy algorithm provides the optimal solution. While the proof for the non-adaptive case is simple, the proof that the adaptive greedy policy is optimal is much more involved and relies on a careful analysis of transitions between system statuses. The proof techniques may be applicable in the analysis of other related problems.

Second, we consider the online setting where the community sizes are unknown in advance, which models the uncertainty about the communities in real applications. We apply the multi-armed bandit (MAB) framework to this task, in which community explorations proceed in multiple rounds, and in each round we explore communities with a budget of $K$, use the feedback to learn about the community size, and adjust the exploration strategy in future rounds. The reward of a round is the the expected number of unique people met in the round. The goal is to maximize the cumulative reward from all rounds, or minimizing the regret, which is defined as the difference in cumulative reward between always using the optimal offline algorithm when knowing the community sizes and using the online learning algorithm. Similar to the offline case, we also consider the non-adaptive and adaptive version of exploration within each round. We provide theoretical regret bounds of $O(\log T)$ for both versions, where $T$ is the number of rounds, which is asymptotically tight. Our analysis uses the special feature of the community exploration problem, which leads to improved coefficients in the regret bounds compared with a simple application of some existing results on combinatorial MABs. Moreover, we also discuss the possibility of using the feedback in previous round to turn the problem into the full information feedback model, which allows us to provide constant regret in this case.

In summary, our contributions include: (a) proposing the study of the community exploration problem to reflect the core of a number of real-world applications; and (b) a systematic study of the problem with rigorous theoretical analysis that covers offline non-adaptive, offline adaptive, online non-adaptive and online adaptive cases, which model the real-world situations of adapting to feedback and handling uncertainty.

## 2    Problem Definition

We model the problem with $m$ disjoint communities $C_1, \ldots, C_m$ with $C = \cup_{i=1}^{m} C_i$, where each community $C_i$ has $d_i$ members. Each exploration (or visit) of one community $C_i$ returns a member of $C_i$ uniformly at random, and we have a total budget of $K$ for explorations. Since we can trivially explore each community once when $K \leq m$, we assume that $K > m$.

We consider both the offline setting where the sizes of the communities $d_1, \ldots, d_m$ are known, and the online setting where the sizes of the communities are unknown. For the offline setting, we further consider two different problems: (1) non-adaptive exploration and (2) adaptive exploration. For the non-adaptive exploration, the explorer needs to predetermine the budget allocation $\boldsymbol{k}$ before the exploration starts, while for the adaptive exploration, she can sequentially select the next community to explore based on previous observations (the members met in the previous community visits). Formally, we use pair $(i, \tau)$ to represent the $\tau$-th exploration of community $C_i$, called an *item*. Let $\mathcal{E} = [m] \times [K]$ be the set of all possible items. A *realization* is a function $\phi \colon \mathcal{E} \to C$ mapping every possible item $(i, \tau)$ to a member in the corresponding community $C_i$, and $\phi(i, \tau)$ represents the member met in the exploration $(i, \tau)$. We use $\Phi$ to denote a random realization, and the randomness comes from the exploration results. From the description above, $\Phi$ follows the distribution such that $\Phi(i, \tau) \in C_i$ is selected uniformly at random from $C_i$ and is independent of all other $\Phi(i', \tau')$'s.

For a budget allocation $\boldsymbol{k} = (k_1, \ldots, k_m)$ and a realization $\phi$, we define the reward $R$ as the number of distinct members met, i.e., $R(\boldsymbol{k}, \phi) = \sum_{i=1}^{m} |\cup_{\tau=1}^{k_i} \{\phi(i, \tau)\}|$, where $|\cdot|$ is the cardinality of the set. The goal of the *non-adaptive exploration* is to find an optimal budget allocation $\boldsymbol{k}^* = (k_1^*, \ldots, k_m^*)$ with given budget $K$, which maximizes the expected reward taken over all possible realizations, i.e.,

$$\boldsymbol{k}^* \in \underset{\boldsymbol{k}\colon \|\boldsymbol{k}\|_1 \leq K}{\arg\max} \ \mathbb{E}_\Phi \left[ R(\boldsymbol{k}, \Phi) \right]. \tag{1}$$

For the adaptive exploration, the explorer sequentially picks a community to explore, meets a random member of the chosen community, then picks the next community, meets another random member of that community, and so on, until the budget is used up. After each selection, the observations so far can be represented as a *partial realization* $\psi$, a function from the subset of $\mathcal{E}$ to $C = \cup_{i=1}^{m} C_i$. Suppose that each community $C_i$ has been explored $k_i$ times. Then the partial realization $\psi$ is a function mapping items in $\cup_{i=1}^{m} \{(i, 1), \ldots, (i, k_i)\}$ (which is also called the domain of $\psi$, denoted as $\mathrm{dom}(\psi)$) to members in communities. The partial realization $\psi$ records the observation on the sequence of explored communities and the members met in this sequence. We say that a partial realization $\psi$ is consistent with realization $\phi$, denoted as $\phi \sim \psi$, if for all item $(i, \tau)$ in the domain of $\psi$, we have $\psi(i, \tau) = \phi(i, \tau)$. The strategy to explore the communities adaptively is encoded as a policy. The policy, denoted as $\pi$, is a function mapping $\psi$ to an item in $\mathcal{E}$, specifying which community to explore next under the partial realization. Define $\pi_K(\phi) = (k_1, \ldots, k_m)$, where $k_i$ is the times the community $C_i$ is explored via policy $\pi$ under realization $\phi$ with budget $K$. More specifically, starting from the partial realization $\psi_0$ with empty domain, for every current partial realization $\psi_s$ at step $s$, policy $\pi$ determines the next community $\pi(\psi_s)$ to explore, meet the member $\phi(\pi(\psi_s))$, such that the new partial realization $\psi_{s+1}$ is adding the mapping from $\pi(\psi_s)$ to $\phi(\pi(\psi_s))$ on top of $\psi_s$. This iteration continues until the communities have been explored $K$ times, and $\pi_K(\phi) = (k_1, \ldots, k_m)$ denotes the resulting exploration vector. The goal of the adaptive exploration is to find an optimal policy $\pi^*$ to maximize the expected adaptive reward, i.e.,

$$\pi^* \in \underset{\pi}{\arg\max} \ \mathbb{E}_\Phi \left[ R(\pi_K(\Phi), \Phi) \right]. \tag{2}$$

We next consider the online setting of community exploration. The learning process proceeds in discrete rounds. Initially, the size of communities $\boldsymbol{d} = (d_1, \ldots, d_m)$ is unknown. In each round $t \geq 1$, the learner needs to determine an allocation or a policy (called an *"action"*) based on the previous-round observations to explore communities (non-adaptively or adaptively). When an action is played, the sets of encountered members for every community are observed as the *feedback* to the player. A learning algorithm $A$ aims to cumulate as much reward (i.e., number of distinct members) as possible by selecting actions properly at each round. The performance of a learning algorithm is measured by the *cumulative regret*. Let $\Phi_t$ be the realization at round $t$. If we explore the communities with predetermined budget allocation in each round, the $T$-round (non-adaptive) regret of a learning algorithm $A$ is defined as

$$\mathrm{Reg}_{\boldsymbol{\mu}}^A(T) = \mathbb{E}_{\Phi_1, \ldots, \Phi_T} \left[ \sum_{t=1}^{T} R(\boldsymbol{k}^*, \Phi_t) - R(\boldsymbol{k}_t^A, \Phi_t) \right], \tag{3}$$

where the budget allocation $\boldsymbol{k}_t^A$ is selected by algorithm $A$ in round $t$. If we explore the communities adaptively in each round, then the $T$-round (adaptive) regret of a learning algorithm $A$ is defined as

$$\mathrm{Reg}_{\boldsymbol{\mu}}^A(T) = \mathbb{E}_{\Phi_1, \ldots, \Phi_T} \left[ \sum_{t=1}^{T} R(\pi_K^*(\Phi_t), \Phi_t) - R(\pi_K^{A,t}(\Phi_t), \Phi_t) \right], \tag{4}$$

where $\pi^{A,t}$ is a policy selected by algorithm $A$ in round $t$. The goal of the learning problem is to design a learning algorithm $A$ which minimizes the regret defined in (3) and (4).

## 3 Offline Optimization for Community Exploration

### 3.1 Non-adaptive Exploration Algorithms

If $C_i$ is explored $k_i$ times, each member in $C_i$ is encountered at least once with probability $1 - (1 - 1/d_i)^{k_i}$. Thus we have $\mathbb{E}_\Phi[|\{\Phi(i, 1), \ldots, \Phi(i, k_i)\}|] = d_i(1 - (1 - 1/d_i)^{k_i})$. Hence $\mathbb{E}_\Phi [R(\boldsymbol{k}, \Phi)]$ is a function of only the budget allocation $\boldsymbol{k}$ and the size $\boldsymbol{d} = (d_1, \ldots, d_m)$ of all communities.

---
**Algorithm 1** Non-Adaptive community exploration with optimal budget allocation
---
1: **procedure** COMMUNITYEXPLORE($\{\mu_1, \ldots, \mu_m\}, K$, non-adaptive)
2:      For $i \in [m], k_i \leftarrow 0$                              ▷ *Line 2-5: budget allocation*
3:      **for** $s = 1, \ldots, K$ **do**
4:          $i^* \leftarrow$ a random elements in $\arg\max_i (1 - \mu_i)^{k_i}$      ▷ *$O(\log m)$ via using priority queue*
5:          $k_{i^*} \leftarrow k_{i^*} + 1$
6:      For $i \in [m]$, explore $C_i$ for $k_i$ times, and put the uniformly met members in multi-set $\mathcal{S}_i$
7: **end procedure**
---

---
**Algorithm 2** Adaptive community exploration with greedy policy
---
1: **procedure** COMMUNITYEXPLORE($\{\mu_1, \ldots, \mu_m\}, K$, adaptive)
2:      For $i \in [m], \mathcal{S}_i \leftarrow \emptyset, c_i \leftarrow 0$      ▷ *Line 2-7: adaptively explore communities with policy $\pi^g$*
3:      **for** $s = 1, \ldots, K$ **do**
4:          $i^* \leftarrow$ a random elements in $\arg\max_i 1 - \mu_i c_i$
5:          $v \leftarrow$ a random member met when $C_{i^*}$ is explored
6:          **if** $v \notin \mathcal{S}_{i^*}$ **then** $c_{i^*} \leftarrow c_{i^*} + 1$                  ▷ *v is not met before*
7:          $\mathcal{S}_{i^*} \leftarrow \mathcal{S}_{i^*} \cup \{v\}$
8: **end procedure**
---

Let $\mu_i = 1/d_i$, and vector $\boldsymbol{\mu} = (1/d_1, \ldots, 1/d_m)$. Henceforth, we treat $\boldsymbol{\mu}$ as the parameter of the problem instance, since it is bounded with $\boldsymbol{\mu} \in [0, 1]^m$. Let $r_{\boldsymbol{k}}(\boldsymbol{\mu}) = \mathbb{E}_\Phi[R(\boldsymbol{k}, \Phi)]$ be the expected reward for the budget allocation $\boldsymbol{k}$. Based on the above discussion, we have

$$r_{\boldsymbol{k}}(\boldsymbol{\mu}) = \sum_{i=1}^{m} d_i (1 - (1 - 1/d_i)^{k_i}) = \sum_{i=1}^{m} (1 - (1 - \mu_i)^{k_i})/\mu_i. \tag{5}$$

Since $k_i$ must be integers, a traditional method like *Lagrange Multipliers* cannot be applied to solve the optimization problem defined in Eq. (1). We propose a *greedy method* consisting of $K$ steps to compute the feasible $\boldsymbol{k}^*$. The greedy method is described in Line 2-5 of Algo. 1.

**Theorem 1.** *The greedy method obtains an optimal budget allocation.*

The time complexity of the greedy method is $O(K \log m)$, which is not efficient for large $K$. We find that starting from the initial allocation $k_i = \left\lceil \frac{(K-m)/\ln(1-\mu_i)}{\sum_{j=1}^{m} 1/\ln(1-\mu_j)} \right\rceil$, the greedy method can find the optimal budget allocation in $O(m \log m)^2$. (See supplementary materials.)

## 3.2    Adaptive Exploration Algorithms

With a slight abuse of notations, we also define $r_\pi(\boldsymbol{\mu}) = \mathbb{E}_\Phi[R(\pi_K(\Phi), \Phi)]$, since the expected reward is the function of the policy $\pi$ and the vector $\boldsymbol{\mu}$. Define $c_i(\psi)$ as the number of distinct members we met in community $C_i$ under partial realization $\psi$. Then $1 - c_i(\psi)/d_i$ is the probability that we can meet a new member in the community $C_i$ if we explore community $C_i$ one more time. A natural approach is to explore community $C_{i^*}$ such that $i^* \in \arg\max_{i \in [m]} 1 - c_i(\psi)/d_i$ when we have partial realization $\psi$. We call such policy as the greedy policy $\pi^g$. The adaptive community exploration with greedy policy is described in Algo. 2. One could show that our reward function is actually an *adaptive submodular* function, for which the greedy policy is guaranteed to achieve at least $(1 - 1/e)$ of the maximized expected reward [13]. However, the following theorem shows that for our community exploration problem, our greedy policy is in fact *optimal*.

**Theorem 2.** *Greedy policy is the optimal policy for our adaptive exploration problem.*

**Proof sketch.** Note that the greedy policy chooses the next community only based on the fraction of unseen members. It does not care which members are already met. Thus, we define $s_i$ as the percentage of members we have not met in a community $C_i$. We introduce the concept of *status*, denoted as $\boldsymbol{s} = (s_1, \ldots, s_m)$. The greedy policy chooses next community based on the current

**Algorithm 3** Combinatorial Lower Confidence Bound (CLCB) algorithm

---
**Input** budget for each round $K$, method (non-adaptive or adaptive)
1: For $i \in [m]$, $T_i \leftarrow 0$ (number of pairs), $X_i \leftarrow 0$ (collision counting), $\hat{\mu}_i \leftarrow 0$ (empirical mean)
2: **for** $t = 1, 2, 3, \ldots$ **do**        ▷ *Line 2-8: online learning*
3:      For $i \in [m]$, $\rho_i \leftarrow \sqrt{\frac{3 \ln t}{2 T_i}}$ ($\rho_i = 0$ if $T_i = 0$)        ▷ *confidence radius*
4:      For $i \in [m]$, $\underline{\mu}_i \leftarrow \max\{0, \hat{\mu}_i - \rho_i\}$        ▷ *lower confidence bound*
5:      $\{\mathcal{S}_1, \ldots, \mathcal{S}_m\} \leftarrow \textsc{CommunityExplore}(\{\underline{\mu}_1, \ldots, \underline{\mu}_m\}, K, \mathsf{method})$   ▷ $\mathcal{S}_i$: *set of met members*
6:      For $i \in [m]$, $T_i \leftarrow T_i + \lfloor |\mathcal{S}_i| / 2 \rfloor$        ▷ *update number of (member) pairs we observe*
7:      For $i \in [m]$, $X_i \leftarrow X_i + \sum_{x=1}^{\lfloor |\mathcal{S}_i| \rfloor / 2} \mathbb{1}\{\mathcal{S}_i[2x-1] = \mathcal{S}_i[2x]\}$     ▷ $\mathcal{S}_i[x]$: *x-th element in* $\mathcal{S}_i$
8:      For $i \in [m]$ and $|\mathcal{S}_i| > 1$, $\hat{\mu}_i \leftarrow X_i / T_i$        ▷ *update empirical mean*

---

status. In the proof, we further extend the definition of reward with a non-decreasing function $f$ as $R(\boldsymbol{k}, \phi) = f\left(\sum_{i=1}^{m} \left| \bigcup_{\tau=1}^{k_i} \{\phi(i, \tau)\} \right|\right)$. Note that the reward function corresponding to the original community exploration problem is simply the identity function $f(x) = x$. Let $F_\pi(\psi, t)$ denote the expected *marginal gain* when we further explore communities for $t$ steps with policy $\pi$ starting from a partial realization $\psi$. We want to prove that for all $\psi$, $t$ and $\pi$, $F_{\pi^g}(\psi, t) \geq F_\pi(\psi, t)$, where $\pi^g$ is the greedy policy and $\pi$ is an arbitrary policy. If so, we simply take $\psi = \emptyset$, and $F_{\pi^g}(\emptyset, t) \geq F_\pi(\emptyset, t)$ for every $\pi$ and $t$ exactly shows that $\pi^g$ is optimal. We prove the above result by an induction on $t$.

Let $C_i$ be the community chosen by $\pi$ based on the partial realization $\psi$. Define $c(\psi) = \sum_i c_i(\psi)$ and $\Delta_{\psi, f} = f(c(\psi) + 1) - f(c(\psi))$. We first claim that $F_{\pi^g}(\psi, 1) \geq F_\pi(\psi, 1)$ holds for all $\psi$ and $\pi$ with the fact that $F_\pi(\psi, 1) = (1 - \mu_i c_i(\psi)) \Delta_{\psi, f}$. Note that the greedy policy $\pi^g$ chooses $C_{i^*}$ with $i^* \in \arg\max_i (1 - \mu_i c_i(\psi))$. Hence, $F_{\pi^g}(\psi, 1) \geq F_\pi(\psi, 1)$.

Next we prove that $F_{\pi^g}(\psi, t+1) \geq F_\pi(\psi, t+1)$ based on the assumption that $F_{\pi^g}(\psi, t') \geq F_\pi(\psi, t')$ holds for all $\psi$, $\pi$, and $t' \leq t$. An important observation is that $F_{\pi^g}(\psi, t)$ has equal value for any partial realization $\psi$ associated with the same status $\boldsymbol{s}$ since the status is enough for the greedy policy to determine the choice of next community. Formally, we define $F_g(\boldsymbol{s}, t) = F_{\pi^g}(\psi, t)$ for any partial realization that satisfies $\boldsymbol{s} = (1 - c_1(\psi)/d_1, \ldots, 1 - c_m(\psi)/d_m)$. Let $C_{i^*}$ denote the community chosen by policy $\pi^g$ under realization $\psi$, i.e., $i^* \in \arg\max_{i \in [m]} 1 - \mu_i c_i(\psi)$. Let $\boldsymbol{I}_i$ be the $m$-dimensional unit vector with one in the $i$-th entry and zeros in all other entries. We show that

$$
\begin{aligned}
F_\pi(\psi, t+1) &\leq c_i(\psi) \cdot \mu_i F_g(\boldsymbol{s}, t) + (d_i - c_i(\psi)) \cdot \mu_i F_g(\boldsymbol{s} - \mu_i \boldsymbol{I}_i, t) + (1 - \mu_i c_i(\psi)) \Delta_{\psi, f} \\
&\leq \mu_{i^*} c_{i^*}(\psi) F_g(\boldsymbol{s}, t) + (1 - \mu_{i^*} c_{i^*}(\psi)) F_g(\boldsymbol{s} - \mu_{i^*} \boldsymbol{I}_{i^*}, t) + (1 - \mu_{i^*} c_{i^*}(\psi)) \Delta_{\psi, f} \\
&= F_g(\boldsymbol{s}, t+1) = F_{\pi^g}(\psi, t+1).
\end{aligned}
$$

The first line is derived directly from the definition and the assumption. The key is to prove the correctness of Line 2 in above inequality. It indicates that if we choose a sub-optimal community at first, and then we switch back to the greedy policy, the expected reward would be smaller. The proof is nontrivial and relies on a careful analysis based on the stochastic transitions among status vectors. We leave detailed analysis in the supplementary materials. Note that the reward function $r_\pi(\boldsymbol{\mu})$ is not necessary adaptive submodular if we extend the reward with the non-decreasing function $f$. Hence, a $(1 - 1/e)$ guarantee for adaptive submodular function [13] is not applicable in this scenario. Our analysis scheme can be applied to any adaptive problems with similar structures.

## 4 Online Learning for Community Exploration

The key of the learning algorithm is to estimate the community sizes. The size estimation problem is defined as inferring unknown set size $d_i$ from random samples obtained via uniformly sampling *with replacement* from the set $C_i$. Various estimators have been proposed [3, 8, 10, 16] for the estimation of $d_i$. The core idea of estimators in [3, 16] are based on "*collision counting*". Let $(u, v)$ be an *unordered pair* of two random elements from $C_i$ and $Y_{u,v}$ be a *pair collision* random variable that takes value 1 if $u = v$ (i.e., $(u, v)$ is *a collision*) and 0 otherwise. It is easy to verify that $\mathbb{E}[Y_{u,v}] = 1/d_i = \mu_i$. Suppose we *independently* take $T_i$ pairs of elements from $C_i$ and $X_i$ of them are collisions. Then $\mathbb{E}[X_i/T_i] = 1/d_i = \mu_i$. The size $d_i$ can be estimated by $T_i/X_i$ (the estimator is valid when $X_i > 0$).

We present our CLCB algorithm in Algorithm 3. In the algorithm, we maintain an unbiased estimation of $\mu_i$ instead of $d_i$ for each community $C_i$ for the following reasons. Firstly, $T_i/X_i$ is not an unbiased estimator of $d_i$ since $\mathbb{E}[T_i/X_i] \geq d_i$ according to the Jensen's inequality. Secondly, the upper confidence bound of $T_i/X_i$ depends on $d_i$, which is unknown in our online learning problem. Thirdly, we need at least $(1+\sqrt{8d_i \ln 1/\delta + 1})/2$ uniformly sampled elements in $C_i$ to make sure that $X_i > 0$ with probability at least $1 - \delta$. We feed the lower confidence bound $\underline{\mu}_i$ to the exploration process since our reward function increases as $\mu_i$ decreases. The idea is similar to CUCB algorithm [7]. The lower confidence bound is small if community $C_i$ is not explored often ($T_i$ is small). Small $\underline{\mu}_i$ motivates us to explore $C_i$ more times. The *feedbacks* after the exploration process at each round are the sets of encountered members $\mathcal{S}_1, \ldots, \mathcal{S}_m$ in communities $C_1, \ldots, C_m$ respectively. Note that for each $i \in [m]$, all pairs of elements in $\mathcal{S}_i$, namely $\{(x,y) \mid x \leq y, x \in \mathcal{S}_i, y \in \mathcal{S}_i \backslash \{x\}\}$ are not mutually independent. Thus, we only use $\lfloor |\mathcal{S}_i|/2 \rfloor$ independent pairs. Therefore, $T_i$ is updated as $T_i + \lfloor |\mathcal{S}_i|/2 \rfloor$ at each round. In each round, the community exploration could either be non-adaptive or adaptive, and the following regret analysis separately discuss these two cases.

## 4.1 Regret Analysis for the Non-adaptive Version

The non-adaptive bandit learning model fits into the general combinatorial multi-armed bandit (CMAB) framework of [7, 20] that deals with nonlinear reward functions. In particular, we can treat the pair collision variable in each community $C_i$ as a base arm, and our expected reward in Eq. (5) is non-linear, and it satisfies the monotonicity and bounded smoothness properties (See Properties 1 and 2). However, directly applying the regret result from [7, 20] will give us an inferior regret bound for two reasons. First, in our setting, in each round we could have multiple sample feedback for each community, meaning that each base arm could be observed multiple times, which is not directly covered by CMAB. Second, to use the regret result in [7, 20], the bounded smoothness property needs to have a bounded smoothness constant independent of the actions, but we can have a better result by using a tighter form of bounded smoothness with action-related coefficients. Therefore, in this section, we provide a better regret result by adapting the regret analysis in [20].

We define the gap $\Delta_{\boldsymbol{k}} = r_{\boldsymbol{k}^*}(\boldsymbol{\mu}) - r_{\boldsymbol{k}}(\boldsymbol{\mu})$ for all action $\boldsymbol{k}$ satisfying $\sum_{i=1}^{m} k_i = K$. For each community $C_i$, we define $\Delta_{\min}^i = \min_{\Delta_{\boldsymbol{k}}>0, k_i>1} \Delta_{\boldsymbol{k}}$ and $\Delta_{\max}^i = \max_{\Delta_{\boldsymbol{k}}>0, k_i>1} \Delta_{\boldsymbol{k}}$. As a convention, if there is no action $\boldsymbol{k}$ with $k_i > 1$ such that $\Delta_{\boldsymbol{k}} > 0$, we define $\Delta_{\min}^i = \infty$ and $\Delta_{\max}^i = 0$. Furthermore, define $\Delta_{\min} = \min_{i \in [m]} \Delta_{\min}^i$ and $\Delta_{\max} = \max_{i \in [m]} \Delta_{\max}^i$. Let $K' = K - m + 1$. We have the regret for Algo. 3 as follows.

**Theorem 3.** *Algo. 3 with non-adaptive exploration method has regret as follows.*

$$Reg_{\boldsymbol{\mu}}(T) \leq \sum_{i=1}^{m} \frac{48\binom{K'}{2}K \ln T}{\Delta_{\min}^i} + 2\binom{K'}{2}m + \frac{\lfloor \frac{K'}{2} \rfloor \pi^2}{3} m\Delta_{\max} = O\left(\sum_{i=1}^{m} \frac{K'^3 \log T}{\Delta_{\min}^i}\right). \quad (6)$$

The proof of the above theorem is an adaption of the proof of Theorem 4 in [20], and the full proof details as well as the detailed comparison with the original CMAB framework result are included in the supplementary materials. We briefly explain our adaption that leads to the regret improvement. We rely on the following monotonicity and 1-norm bounded smoothness properties of our expected reward function $r_{\boldsymbol{k}}(\boldsymbol{\mu})$, similar to the ones in [7, 20].

**Property 1** (Monotonicity). *The reward function $r_{\boldsymbol{k}}(\boldsymbol{\mu})$ is monotonically decreasing, i.e., for any two vectors $\boldsymbol{\mu} = (\mu_1, \ldots, \mu_m)$ and $\boldsymbol{\mu}' = (\mu_1', \ldots, \mu_m')$, we have $r_{\boldsymbol{k}}(\boldsymbol{\mu}) \geq r_{\boldsymbol{k}}(\boldsymbol{\mu}')$ if $\mu_i \leq \mu_i' \; \forall i \in [m]$.*

**Property 2** (1-Norm Bounded Smoothness). *The reward function $r_{\boldsymbol{k}}(\boldsymbol{\mu})$ satisfies the 1-norm bounded smoothness property, i.e., for any two vectors $\boldsymbol{\mu} = (\mu_1, \cdots, \mu_m)$ and $\boldsymbol{\mu}' = (\mu_1', \cdots, \mu_m')$, we have $|r_{\boldsymbol{k}}(\boldsymbol{\mu}) - r_{\boldsymbol{k}}(\boldsymbol{\mu}')| \leq \sum_{i=1}^{m} \binom{k_i}{2}|\mu_i - \mu_i'| \leq \binom{K'}{2} \sum_{i=1}^{m} |\mu_i - \mu_i'|$.*

We remark that if we directly apply the CMAB regret bound of Theorem 4 in [20], we need to revise the update procedure in Lines 6-8 of Algo. 3 so that each round we only update one observation for each community $C_i$ if $|\mathcal{S}_i| > 1$. Then we would obtain a regret bound $O\left(\sum_i \frac{K'^4 m \log T}{\Delta_{\min}^i}\right)$, which means that our regret bound in Eq. (6) has an improvement of $O(K'm)$. This improvement is exactly due to the reason we give earlier, as we now explain with more details.

For all the random variables introduced in Algo. 3, we add the subscript $t$ to denote their value at the *end* of round $t$. For example, $T_{i,t}$ is the value of $T_i$ at the end of round $t$. First, the improvement of

the factor $m$ comes from the use of a tighter bounded smoothness in Property 2, namely, we use the bound $\sum_{i=1}^{m} \binom{k_i}{2}|\mu_i - \mu_i'|$ instead of $\binom{K'}{2}\sum_{i=1}^{m}|\mu_i - \mu_i'|$. The CMAB framework in [20] requires the bounded smoothness constant to be independent of actions. So to apply Theorem 4 in [20], we have to use the bound $\binom{K'}{2}\sum_{i=1}^{m}|\mu_i - \mu_i'|$. However, in our case, when using bound $\sum_{i=1}^{m}\binom{k_i}{2}|\mu_i - \mu_i'|$, we are able to utilize the fact $\sum_{i=1}^{m}\binom{k_i}{2} \leq \binom{K'}{2}$ to improve the result by a factor of $m$. Second, the improvement of the $O(K')$ factor, more precisely a factor of $(K'-1)/2$, is achieved by utilizing multiple feedback in a single round and a more careful analysis of the regret utilizing the property of the right Riemann summation. Specifically, let $\Delta_{\boldsymbol{k}_t} = r_{\boldsymbol{k}^*}(\boldsymbol{\mu}) - r_{\boldsymbol{k}_t}(\boldsymbol{\mu})$ be the reward gap. When the estimate is within the confidence radius, we have $\Delta_{\boldsymbol{k}_t} \leq \sum_{i=1}^{m} \frac{c(k_{i,t}-1)}{2}/\sqrt{T_{i,t-1}} \leq c\sum_{i=1}^{m} \lfloor k_{i,t}/2 \rfloor/\sqrt{T_{i,t-1}}$, where $c$ is a constant. In Algo. 3, we have $T_{i,t} = T_{i,t-1} + \lfloor k_{i,t}/2 \rfloor$ because we allow multiple feedback in a single round. Then $\sum_{t \geq 1, T_{i,t} \leq L_i(T)} \lfloor k_{i,t}/2 \rfloor/\sqrt{T_{i,t-1}}$ is the form of a right Riemann summation, which achieves the maximum value when $k_{i,t} = K'$. Here $L_i(T)$ is a $\ln T$ function with some constants related with community $C_i$. Hence the regret bound $\sum_{t=1}^{T} \Delta_{\boldsymbol{k}_t} \leq c\sum_{i=1}^{m} \sum_{t \geq 1, T_{i,t} \leq L_i(T)} \lfloor \frac{k_{i,t}}{2} \rfloor/\sqrt{T_{i,t-1}} \leq 2c\sum_{i=1}^{m} \sqrt{L_i(T)}$. However, if we use the original CMAB framework, we need to set $T_{i,t} = T_{i,t-1} + \mathbb{1}\{k_{i,t} > 1\}$. In this case, we can only bound the regret as $\sum_{t=1}^{T} \Delta_{\boldsymbol{k}_t} = c\sum_{i=1}^{m} \sum_{t \geq 1, T_{i,t} \leq L_i(T)} (k_{i,t} - 1)/2\sqrt{T_{i,t-1}} \leq 2c\frac{K'-1}{2}\sum_{i=1}^{m} \sqrt{L_i(T)}$, leading to an extra factor of $(K'-1)/2$.

**Justification for Algo. 3.** In Algo. 3, we only use the members in current round to update the estimator. This is practical for the situation where the member identifiers are changing in different rounds for privacy protection. Privacy gains much attention these days. Consider the online advertising scenario we explain in the introduction. Whenever a user clicks an advertisement, the advertiser would store the user information (e.g. Facebook ID, IP address etc.) to identify the user and correlated with past visits of the user. If such user identifiers are fixed and do not change, the advertiser could easily track user behavior, which may result in privacy leak. A reasonable protection for users is to periodically change user IDs (e.g. Facebook can periodically change user hash IDs, or users adopt dynamic IP addresses, etc.), so that it is difficult for the advertiser to track the same user over a long period of time. Under such situation, it may be likely that our learning algorithm can still detect ID collisions within the short period of each learning round, but cross different rounds, collisions may not be detectable due to ID changes.

**Full information feedback.** Now we consider the scenario where the member identifiers are fixed over all rounds, and design an algorithm with a constant regret bound. Our idea is to ensure that we can observe at least one pair of members in every community $C_i$ in each round $t$. We call such guarantee as *full information feedback*. If we only use members revealed in current round, we cannot achieve this goal since we have no observation of new pairs for a community $C_i$ when $k_i = 1$. To achieve full information feedback, we use at least one sample from the previous round to form a pair with a sample in the current round to generate a valid pair collision observation. In particular, we revise the Line 3, 6, and 7 as follows. Here we use $u_0$ to represent the last member in $\mathcal{S}_i$ in the previous round (let $u_0 = $ null when $t = 1$) and $u_x (x > 0)$ to represent the $x$-th members in $\mathcal{S}_i$ in the current round. The revision of Line 3 implies that we use the empirical mean $\hat{\mu}_i = X_i/T_i$ instead of the lower confidence bound in the function COMMUNITYEXPLORE.

Line 3:  For $i \in [m], \rho_i = 0$;  Line 6:  For $i \in [m], T_i \leftarrow T_i + |\mathcal{S}_i| - \mathbb{1}\{t = 1\}$,

$$\text{Line 7:}\quad \text{For } i \in [m], X_i \leftarrow X_i + \sum_{x=0}^{|\mathcal{S}_i|-1} \mathbb{1}\{u_x = u_{x+1}\}. \tag{7}$$

**Theorem 4.** *With the full information feedback revision in Eq. (7), Algo. 3 with non-adaptive exploration method has a constant regret bound. Specifically,*

$$Reg_{\boldsymbol{\mu}}(T) \leq \left(2 + 2me^2 K'^2 (K'-1)^2/\Delta_{\min}^2\right) \Delta_{\max}.$$

Note that we cannot apply the Hoeffding bound in [14] directly since the random variables $\mathbb{1}\{u_x = u_{x+1}\}$ we obtain during the online learning process are not mutually independent. Instead, we apply a concentration bound in [9] that is applicable to variables that have local dependence relationship.

## 4.2 Regret Analysis for the Adaptive Version

For the adaptive version, we feed the lower confidence bound $\underline{\mu}_t$ into the adaptive community exploration procedure, namely $\textsc{CommunityExplore}(\{\underline{\mu}_1, \ldots, \underline{\mu}_m\}, K, \text{adaptive})$ in round $t$. We denote the policy implemented by this procedure as $\pi^t$. Note that both $\pi^g$ and $\pi^t$ are based on the greedy procedure $\textsc{CommunityExplore}(\cdot, K, \text{adaptive})$. The difference is that $\pi^g$ uses the true parameter $\boldsymbol{\mu}$ while $\pi^t$ uses the lower bound parameter $\underline{\mu}_t$. More specifically, given a partial realization $\psi$, the community chosen by $\pi^t$ is $C_{i^*}$ where $i^* \in \arg\max_{i \in [m]} 1 - c_i(\psi)\underline{\mu}_{i,t}$. Recall that $c_i(\psi)$ is the number of distinct encountered members in community $C_i$ under partial realization $\psi$.

We first properly define the metrics $\Delta_{\min}^{i,k}$ and $\Delta_{\max}^{(k)}$ used in the regret bound as follows. Consider a specific full realization $\phi$ where $\{\phi(i, 1), \ldots, \phi(i, d_i)\}$ are $d_i$ distinct members in $C_i$ for $i \in [m]$. The realization $\phi$ indicates that we will obtain a new member in the first $d_i$ exploration of community $C_i$. Let $U_{i,k}$ denote the number of times community $C_i$ is selected by policy $\pi^g$ in the first $k - 1(k > m)$ steps under the special full realization $\phi$ we define previously. We define $\Delta_{\min}^{i,k} = (\mu_i U_{i,k} - \min_{j \in [m]} \mu_j U_{j,k})/U_{i,k}$. Conceptually, the value $\mu_i U_{i,k} - \min_{j \in [m]} \mu_j U_{j,k}$ is gap in the expected reward of the next step between selecting a community by $\pi^g$ (the optimal policy) and selecting community $C_i$, when we already meet $U_{j,k}$ distinct members in $C_j$ for $j \in [m]$. When $\mu_i U_{i,k} = \min_{j \in [m]} \mu_j U_{j,k}$, we define $\Delta_{\min}^{i,k} = \infty$. Let $\pi$ be another policy that chooses the same sequence of communities as $\pi^g$ when the number of met members in $C_i$ is no more than $U_{i,k}$ for all $i \in [m]$. Note that policy $\pi$ chooses the same communities as $\pi^g$ in the first $k - 1$ steps under the special full realization $\phi$. Actually, the policy $\pi$ is the same as $\pi^g$ for at least $k - 1$ steps. We use $\Pi_k$ to denote the set of all such policies. We define $\Delta_{\max}^{(k)}$ as the maximum reward gap between the policy $\pi \in \Pi_k$ and the optimal policy $\pi^g$, i.e., $\Delta_{\max}^{(k)} = \max_{\pi \in \Pi_k} r_{\pi^g}(\boldsymbol{\mu}) - r_\pi(\boldsymbol{\mu})$. Let $D = \sum_{i=1}^m d_i$.

**Theorem 5.** *Algo. 3 with adaptive exploration method has regret as follows.*

$$Reg_{\boldsymbol{\mu}}(T) \leq \left( \sum_{i=1}^m \sum_{k=m+1}^{\min\{K,D\}} \frac{6\Delta_{\max}^{(k)}}{(\Delta_{\min}^{i,k})^2} \right) \ln T + \frac{\lfloor \frac{K'}{2} \rfloor \pi^2}{3} \sum_{i=1}^m \sum_{k=m+1}^{\min\{K,D\}} \Delta_{\max}^{(k)}. \tag{8}$$

**Theorem 6.** *With the full information feedback revision in Eq. (7), Algo. 3 with adaptive exploration method has a constant regret bound. Specifically,*

$$Reg_{\boldsymbol{\mu}}(T) \leq \sum_{i=1}^m \sum_{k=m+1}^{\min\{K,D\}} \left( 2/\varepsilon_{i,k}^4 + 1 \right) \Delta_{\max}^{(k)}.$$

*where $\varepsilon_{i,k}$ is defined as (here $i_k^* \in \arg\min_{i \in [m]} \mu_i U_{i,k}$)*

$$\varepsilon_{i,k} \triangleq (\mu_i U_{i,k} - \mu_{i_k^*} U_{i_k^*,k})/(U_{i,k} + U_{i_k^*,k}) \text{ for } i \neq i_k^* \text{ and } \varepsilon_{i,k} = \infty \text{ for } i = i_k^*.$$

Gabillon et al. [11] analyzes a general adaptive submodular function maximization in bandit setting. We have a regret bound in similar form as (8) if we directly apply Theorem 1 in [11]. However, their version of $\Delta_{\max}^{(k)}$ is an upper bound on the expected reward of policy $\pi^g$ from $k$ steps forward, which is larger than our $\Delta_{\max}^{(k)}$. Their version of $\Delta_{\min}^{i,k}$ is the minimum $(\mu_i c_i(\psi) - \min_{j \in [m]} \mu_j c_j(\psi))/c_i(\psi)$ for all partial realization $\psi$ obtained after policy $\pi^g$ is executed for $k$ steps, which is smaller than our $\Delta_{\min}^{i,k}$. Our regret analysis is based on counting how many times $\pi^g$ and $\pi^t$ choose different communities under the special full realization $\phi$, while the analysis in [11] is based on counting how many times $\pi^g$ and $\pi^t$ choose different communities under all possible full realizations.

**Discussion.** In this paper, we consider the online learning problem that consists of $T$ rounds, and during each round, we explore the communities with a budget $K$. Our goal is to maximize the *cumulative reward* in $T$ rounds. Another important and natural setting is described as follows. We start to explore communities with unknown sizes, and update the parameters every time we explore the community for *one step* (or for a few steps). Different from the setting defined in this paper, here *a member will not contribute to the reward if it has been met in previous rounds*. To differentiate the two settings, let's call the latter one the "*interactive community exploration*", while the former one the "*repeated community exploration*". Both the repeated community exploration defined in this paper and the interactive community exploration we will study as the future work have corresponding applications. The former is suitable for online advertising where in each round the advertiser promotes different products. Hence the rewards in different rounds are additive. The latter corresponds to the adaptive online advertising for the same product, and thus the rewards in different rounds are dependent.

# 5 Related Work

Golovin and Krause [13] show that a greedy policy could achieve at least $(1 - 1/e)$ approximation for the adaptive submodular function. The result could be applied to our offline adaptive problem, but by an independent analysis we show the better result that the greedy policy is optimal. Multi-armed bandit (MAB) problem is initiated by Robbins [18] and extensively studied in [2, 4, 19]. Our online learning algorithm is based on the extensively studied *Upper Confidence Bound* approach [1]. The non-adaptive community exploration problem in the online setting fits into the general combinatorial multi-armed bandit (CMAB) framework [6, 7, 12, 17, 20], where the reward is a set function of base arms. The CMAB problem is first studied in [12], and its regret bound is improved by [7, 17]. We leverage the analysis framework in [7, 20] and prove a tighter bound for our algorithm. Gabillon et al. [11] define an adaptive submodular maximization problem in bandit setting. Our online adaptive exploration problem is a instance of the problem defined in [11]. We prove a tighter bound than the one in [11] by using the properties of our problem.

Our model bears similarities to the optimal discovery problem proposed in [5] such as we both have disjoint assumption, and both try to maximize the number of target elements. However, there are also some differences: (a) We use different estimators for our critical parameters, because our problem setting is different. (b) Their online model is closer to the interactive community exploration we explained in 4.2 , while our online model is on repeated community exploration. As explained in 4.2, the two online models serve different applications and have different algorithms and analyses. (c) We also have more comprehensive studies on the offline cases.

# 6 Future Work

In this paper, we systematically study the community exploration problems. In the offline setting, we propose the greedy methods for both of non-adaptive and adaptive exploration problems. The optimality of the greedy methods are rigorously proved. We also analyze the online setting where the community sizes are unknown initially. We provide a CLCB algorithm for the online community exploration. The algorithm has $O(\log T)$ regret bound. If we further allow the full information feedback, the CLCB algorithm with some minor revisions has a constant regret.

Our study opens up a number of possible future directions. For example, we can consider various extensions to the problem model, such as more complicated distributions of member meeting probabilities, overlapping communities, or even graph structures between communities. We could also study the gap between non-adaptive and adaptive solutions.

**Acknowledgments**

We thank Jing Yu from School of Mathematical Sciences at Fudan University for her insightful discussion on the offline problems, especially, we thank Jing Yu for her method to find a good initial allocation, which leads to a faster greedy method. Wei Chen is partially supported by the National Natural Science Foundation of China (Grant No. 61433014). The work of John C.S. Lui is supported in part by the GRF Grant 14208816.

## Footnotes

[1]The model can be extended to meet multiple members per visit, but for simplicity, we consider meeting one member per visit in this paper.

[2]We thank Jing Yu from School of Mathematical Sciences at Fudan University for her method to find a good initial allocation, which leads to a faster greedy method.

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
