[Supplementary Material]

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

Similar to $\pi^g$, the policy $\pi^t$ also chooses next community to explore based on current *status*. Let $\boldsymbol{s} = (s_1, \ldots, s_m) = (1 - c_1(\psi)\mu_1, \ldots, 1 - c_m(\psi)\mu_m)$ be the corresponding status to the partial realization $\psi$. Here $s_i$ is the percentage of unmet members in the community $C_i$. For any partial realization $\psi$ having status $\boldsymbol{s}$, the policy $\pi^t$ choose $C_{i^*}$ to explore, where $i^* \in \arg\max_{i \in [m]}(\underline{\mu}_{i,t}/\mu_i)s_i + (\mu_i - \underline{\mu}_i)/\mu_i$. When $\underline{\mu}_{i,t} \leq \mu_i$, we have $(\underline{\mu}_{i,t}/\mu_i)s_i + (\mu_i - \underline{\mu}_i)/\mu_i \geq s_i$, which means that the percentage of unmet members in $C_i$ is overestimated by $\pi^t$.

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

# Supplementary Materials

## A  Improved Budget Allocation Algorithm

**Theorem 1.** *The greedy method obtains an optimal budget allocation.*

**Proof.** Let $r_i(j) = \mathbb{E}_\Phi[|\{\Phi(i,1), \dots, \Phi(i,j)\}|] = d_i(1 - (1 - 1/d_i)^j)$ denote the expected reward when the community $i$ is explored $j$ times. Then we have that the marginal gain $r_i(j+1) - r_i(j) = (1 - \mu_i)^j$. Define a matrix $\boldsymbol{X} \in \mathbb{R}^{m \times K}$, where the $(i,j)$-th entry $X_{i,j}$ is $(1 - \mu_i)^{j-1}$. When the budget allocation is $\boldsymbol{k} = (k_1, \dots, k_m)$, the expected reward $r_{\boldsymbol{k}}(\boldsymbol{\mu})$ can be written as the sum of elements in $\boldsymbol{X}$, i.e., $r_{\boldsymbol{k}}(\boldsymbol{\mu}) = \sum_{i=1}^m \sum_{j=1}^{k_i} X_{i,j}$. A key property of $\boldsymbol{X}$ is that the value in each row is decreasing with respect to the column index $j$. Hence, for every $s \geq 1$, the $s$-th step of the greedy method chooses the $s$-th largest value in $\boldsymbol{X}$. At step $s = K$, the greedy method finds the largest $K$ values in matrix $\boldsymbol{X}$. We can conclude that the greedy method obtains a budget allocation that maximizes the reward $r_{\boldsymbol{k}}(\boldsymbol{\mu})$. ∎

We propose a budget allocation algorithm which has time complexity $O(m \log m)$ in Algo. 4. The basic idea is to find a good initial allocation that is not far from the optimal allocation. Then starting from the initial allocation, we run our original greedy method.

---
**Algorithm 4** Budget allocation algorithm

---
**Input** parameters $\boldsymbol{\mu}$, budget $K > m$
1: For $i \in [m]$, $k_i = \lceil ((K - m)/\ln(1 - \mu_i))/(\sum_{j=1}^m 1/\ln(1 - \mu_j)) \rceil$ ▷ *A good initial allocation*
2: **while** $\sum_{i=1}^m k_i < k$ **do**
3:    $i^* \leftarrow \arg\max_i (1 - \mu_i)^{k_i}$ ▷ *$O(\log m)$ via using priority queue*
4:    $k_{i^*} \leftarrow k_{i^*} + 1$

---

**Lemma 1** (Basic property of optimal allocation). *Let $\boldsymbol{k}^*$ be the optimal budget allocation when the parameter of the community is $\boldsymbol{\mu}$. For $i, j \in [m]$, we have*

$$(1 - \mu_i)^{(k_i^* - 1)} \geq (1 - \mu_j)^{k_j^*}.$$

**Proof.** We define budget allocation $\boldsymbol{k}'$ which is the same as $\boldsymbol{k}^*$ except that $k_i' = k_i^* - 1$ and $k_j' = k_j^* + 1$. If $(1 - \mu_i)^{(k_i^* - 1)} < (1 - \mu_j)^{k_j^*}$ and $i \neq j$, then we have

$$r_{\boldsymbol{k}'}(\boldsymbol{\mu}) = r_{\boldsymbol{k}^*}(\boldsymbol{\mu}) - (1 - \mu_i)^{(k_i^* - 1)} + (1 - \mu_j)^{k_j^*} > r_{\boldsymbol{k}^*}(\boldsymbol{\mu}),$$

which is contradict with the fact that $\boldsymbol{k}^*$ is the optimal solution. This proves the lemma. ∎

**Lemma 2** (Allocation lower bound). *Let $\boldsymbol{k}^*$ be the optimal budget allocation when the parameter of the communities is $\boldsymbol{\mu}$. Define $\boldsymbol{k}^- = (k_1^-, \dots, k_m^-)$ where*

$$k_i^- = \frac{(K - m)/\ln(1 - \mu_i)}{\sum_{j=1}^m 1/\ln(1 - \mu_j)}.$$

*We have $k_i^* \geq k_i^-$.*

**Proof.** According to the definition of $\boldsymbol{k}^-$, we have $k_i^- \ln(1 - \mu_i) = k_j^- \ln(1 - \mu_j)$ for $i, j \in [m]$. If we can find $i$ such that $k_i^- + 1 \leq k_i^*$, then

$$(1 - \mu_j)^{k_j^-} = (1 - \mu_i)^{k_i^-} \geq (1 - \mu_i)^{k_i^* - 1} \geq (1 - \mu_j)^{k_j^*}.$$

Hence $k_j^- \leq k_j^*$. On the other hand, we can always find $k_i^- + 1 \leq k_i^*$ since $\sum_{i=1}^m (k_i^- + 1) = K$. ∎

In Algo. 4, we start with the lower bound $\boldsymbol{k}^-$ of the optimal allocation. Since $\sum_{i=1}^m k_i^- = K - m$, we have $\sum_{i=1}^m |\lceil k_i^- \rceil - k_i^*| \leq \sum_{i=1}^m |k_i^- - k_i^*| = m$, which indicates Algo. 4 obtains the optimal budget allocation within $m$ steps. We also provide an upper bound $\boldsymbol{k}^+$ in the following. The upper bound is also close to the optimal budget since $\sum_{i=1}^m |\lfloor k_i^+ \rfloor - k_i^*| \leq \sum_{i=1}^m |k_i^+ - k_i^*| = m$.

**Lemma 3** (Allocation upper bound). *Let $\boldsymbol{k}^*$ be the optimal budget allocation when the parameter of the communities is $\boldsymbol{\mu}$. Define $\boldsymbol{k}^+ = (k_1^+, \ldots, k_m^+)$ where*

$$k_i^+ = \frac{K/\ln(1 - \mu_i)}{\sum_{j=1}^m 1/\ln(1 - \mu_j)} + 1.$$

*We have $k_i^* \leq k_i^+$.*

**Proof.** According to the definition of $\boldsymbol{k}^+$, we have $(k_i^+ - 1)\ln(1 - \mu_i) = (k_j^+ - 1)\ln(1 - \mu_j)$ for $i, j \in [m]$. If we can find $i$ such that $k_i^+ - 1 \geq k_i^*$, then

$$(1 - \mu_j)^{k_j^+ - 1} = (1 - \mu_i)^{k_i^+ - 1} \leq (1 - \mu_i)^{k_i^*} \leq (1 - \mu_j)^{k_j^* - 1}.$$

Hence $k_j^+ \geq k_j^*$. On the other hand, we can always find $k_i^+ - 1 \geq k_i^*$ since $\sum_{i=1}^m (k_i^+ - 1) = K$. ∎

# B Properties of Greedy Policy

In the following, we show some important properties of the greedy policy. We further extend the definition of reward with a non-decreasing function $f$ as $R(\boldsymbol{k}, \phi) = f\left(\sum_{i=1}^m \left|\bigcup_{\tau=1}^{k_i}\{\phi(i, \tau)\}\right|\right)$.

## B.1 Optimality of greedy policy

In this part, we prove that the greedy policy is the optimal policy for our adaptive community exploration problem. To prove the optimality, we first rewrite the proof sketch of Theorem 2, and then provide the supporting Lemma 4&5.

**Theorem 2.** *Greedy policy is the optimal policy for our adaptive exploration problem.*

**Proof.** Let $F_\pi(\psi, t)$ denote the expected *marginal gain* when we further explore communities for $t$ steps with policy $\pi$ starting from a partial realization $\psi$. We want to prove that for all $\psi$, $t$ and $\pi$, $F_{\pi^g}(\psi, t) \geq F_\pi(\psi, t)$, where $\pi^g$ is the greedy policy and $\pi$ is an arbitrary policy. If so, we simply take $\psi = \emptyset$, and $F_{\pi^g}(\emptyset, t) \geq F_\pi(\emptyset, t)$ for every $\pi$ and $t$ exactly shows that $\pi^g$ is optimal. We prove the above result by an induction on $t$. Recall that $c_i(\psi)$ is the number of distinct members met in community $C_i$ under the partial realization $\psi$. Define $c(\psi) = \sum_i c_i(\psi)$ and $\Delta_{\psi, f} = f(c(\psi) + 1) - f(c(\psi))$.

For all $\psi$ and $\pi$, we first claim that $F_{\pi^g}(\psi, 1) \geq F_\pi(\psi, 1)$ holds. Suppose that policy $\pi$ chooses community $C_i$ to explore based on the partial realization $\psi$. Since the exploration will return a new member with probability $1 - \mu_i c_i(\psi)$, the expected marginal gain $F_\pi(\psi, 1)$ is $(1 - \mu_i c_i(\psi))[f(c(\psi) + 1) - f(c(\psi))]$. Note that the greedy policy $\pi^g$ chooses community $C_{i^*}$ to explore with $i^* \in \arg\max_j (1 - \mu_j c_j(\psi))$, and $\Delta_{\psi, f}$ does not depend on the policy. Hence, $F_{\pi^g}(\psi, 1) \geq F_\pi(\psi, 1)$.

Assume $F_{\pi^g}(\psi, t') \geq F_\pi(\psi, t')$ holds for all $\psi$, $\pi$, and $t' \leq t$. Our goal is to prove that $F_{\pi^g}(\psi, t + 1) \geq F_\pi(\psi, t + 1)$. Suppose that in the first step after $\psi$, policy $\pi$ chooses $C_i$ to explore based on partial realization $\psi$, and let $\pi(\psi) = (i, \tau)$. Define $E_\psi$ as the event that the member $\Phi(i, \tau)$ is not met in partial realization $\psi$, for $\Phi \sim \psi$. In the following, we represent partial realization $\psi$ equivalently as a relation $\{((i, \tau), \psi(i, \tau)) \mid (i, \tau) \in \text{dom}(\psi)\}$, so we could use $\psi \cup \{((i, \tau), \Phi(i, \tau))\}$ to represent the new partial realization extended from $\psi$ by one step with $(i, \tau)$ added to the domain and $\Phi(i, \tau)$ as the member met for this exploration of $C_i$. Then we have

$$F_\pi(\psi, t + 1) = \sum_{v \in C_i} \Pr(\Phi(i, \tau) = v)\, \mathbb{E}_\Phi[F_\pi(\psi, t + 1) \mid \Phi \sim \psi, \Phi(i, \tau) = v]$$

$$= \sum_{v \in C_i} \mu_i \mathbb{E}_\Phi[F_\pi(\psi \cup \{((i, \tau), \Phi(i, \tau))\}, t) + f(c(\psi) + \mathbb{1}\{E_\psi\}) - f(c(\psi)) \mid \Phi \sim \psi, \Phi(i, \tau) = v]$$

$$\leq \sum_{v \in C_i} \mu_i \mathbb{E}_\Phi[F_{\pi^g}(\psi \cup \{((i, \tau), \Phi(i, \tau))\}, t) \mid \Phi \sim \psi, \Phi(i, \tau) = v] + (1 - \mu_i c_i(\psi))\Delta_{\psi, f}.$$

The 2nd line above is derived directly from the definition of $F_\pi(\psi, t)$. The 3rd line is based on the induction hypothesis that $F_\pi(\psi', t) \leq F_{\pi^g}(\psi', t)$ holds for all $\psi'$. An important observation is that $F_{\pi^g}(\psi, t)$ has equal value for any partial realization $\psi$ associated with the same status $\boldsymbol{s}$ since the

status is enough for the greedy policy to determine the choice of next community. Formally, we define $F_g(\boldsymbol{s}, t) = F_{\pi^g}(\psi, t)$ for any partial realization that satisfies $\boldsymbol{s} = (1 - c_1(\psi)/d_1, \ldots, 1 - c_m(\psi)/d_m)$. Let $C_{i^*}$ denote the community chosen by policy $\pi^g$ under realization $\psi$, i.e., $i^* \in \arg\max_{i \in [m]} 1 - c_i(\psi)\mu_i$. Let $\boldsymbol{I}_i$ be the $m$-dimensional unit vector with 1 in the $i$-th entry and 0 in all other entries. Therefore,

$$
\begin{aligned}
F_\pi(\psi, t + 1) &\leq c_i(\psi) \cdot \mu_i F_g(\boldsymbol{s}, t) + (d_i - c_i(\psi)) \cdot \mu_i F_g(\boldsymbol{s} - \mu_i \boldsymbol{I}_i, t) + (1 - \mu_i c_i(\psi))\Delta_{\psi, f} \\
&\leq \mu_{i^*} c_{i^*}(\psi) F_g(\boldsymbol{s}, t) + (1 - \mu_{i^*} c_{i^*}(\psi)) F_g(\boldsymbol{s} - \mu_{i^*} \boldsymbol{I}_{i^*}, t) + (1 - \mu_{i^*} c_{i^*}(\psi))\Delta_{\psi, f} \\
&\quad\quad\quad\quad\quad\quad\quad\quad\quad\quad\quad\quad\quad\quad\quad\quad\quad\quad\quad\quad\quad\quad\quad\quad \text{(Lemma 5)} \\
&= F_g(\boldsymbol{s}, t + 1) = F_{\pi^g}(\psi, t + 1). \quad\quad\quad\quad\quad\quad\quad\quad\quad\quad\quad \text{(Lemma 4)}
\end{aligned}
$$

The key is to prove the correctness of Line 2 in above equation. It indicates that if we choose a sub-optimal community at first, and then we switch back to the greedy policy, the expected reward would be smaller. The proof is nontrivial and relies on a careful analysis based on the stochastic transitions among status vectors. The above result completes the induction step for $t + 1$. Thus the theorem holds. ∎

**Lemma 4.** *Let $\boldsymbol{s} = (s_1, \ldots, s_m)$ be a status where each entry $s_i \in [0, 1]$. We have*

$$
F_g(\boldsymbol{s}, t + 1) = (1 - s_{i^*})F_g(\boldsymbol{s}, t) + s_{i^*} F_g(\boldsymbol{s} - \mu_{i^*} \boldsymbol{I}_{i^*}, t) + s_{i^*}(f(c(\psi) + 1) - f(c(\psi))),
$$

*where $i^* = \arg\max_{i \in [m]} s_i$. Here $\psi$ is any partial realization corresponding to status $\boldsymbol{s}$.*

**Proof.** For any partial realization $\psi$ associated with status $\boldsymbol{s}$, $\pi^g$ would choose community $i^*$. With probability $\mu_{i^*} c_{i^*}(\psi) = 1 - s_{i^*}$, we will obtain a member that is already met. If so, the communities stay at the same status. Hence, with probability $1 - s_{i^*}$, the expected extra reward is $F_g(\boldsymbol{s}, t)$ after the first step exploration. With probability $1 - \mu_{i^*} c_{i^*}(\psi) = s_{i^*}$, we will obtain an unseen member in $C_{i^*}$. The communities will transit to next status $\boldsymbol{s} - \mu_{i^*} \boldsymbol{I}_{i^*}$. Therefore, with probability $s_{i^*}$, the expected extra reward is $F_g(\boldsymbol{s} - \boldsymbol{I}_{i^*}, t) + f(c(\psi) + 1) f(c(\psi))$ after the first step exploration. ∎

**Lemma 5.** *Let $\boldsymbol{s} = (s_1, \ldots, s_m)$ be a status where each entry $s_i \in [0, 1]$ and $\psi$ be any partial realization corresponding to $\boldsymbol{s}$. We have*

$$
\begin{aligned}
&(1 - s_i)F_g(\boldsymbol{s}, t) + s_i F_g(\boldsymbol{s} - \mu_i \boldsymbol{I}_i, t) + s_i \Delta_c \\
&\leq (1 - s_{i^*})F_g(\boldsymbol{s}, t) + s_{i^*} F_g(\boldsymbol{s} - \mu_{i^*} \boldsymbol{I}_{i^*}, t) + s_{i^*} \Delta_c,
\end{aligned} \quad\quad (9)
$$

*where $i^* \in \arg\max_{i \in [m]} s_i$, $s_i < s_{i^*}$ and $\Delta_c = f(c(\psi) + 1) - f(c(\psi))$.*

**Proof.** Let $A(\boldsymbol{s}, i, t)$ denote the first line of Eq. (9), i.e.,

$$
A(\boldsymbol{s}, i, t) = (1 - s_i)F_g(\boldsymbol{s}, t) + s_i F_g(\boldsymbol{s} - \mu_i \boldsymbol{I}_i, t) + s_i \Delta_c.
$$

Note that $A(\boldsymbol{s}, i, t)$ is the expected reward of the following adaptive process.

1. At the first step, choose an arbitrary community $C_i$ (different from $C_{i^*}$) to explore.

2. From the second step to the $(t + 1)$-th step, explore communities with the greedy policy $\pi^g$.

Similarly, $A(\boldsymbol{s}, i^*, t)$ is the expected reward of the $t + 1$ step community exploration via the greedy policy, i.e., $A(\boldsymbol{s}, i^*, t) = F_g(\boldsymbol{s}, t + 1)$. Eq. (9) can be written as $A(\boldsymbol{s}, i, t) \leq F_g(\boldsymbol{s}, t + 1)$. We prove this inequality by induction. When $t = 0$, we have $A(\boldsymbol{s}, i, t) = s_i \Delta_c$, and $A(\boldsymbol{s}, i^*, t) = s_{i^*} \Delta_c$. Hence, $A(\boldsymbol{s}, i, t) \leq A(\boldsymbol{s}, i^*, t) = F_g(\boldsymbol{s}, t + 1)$ when $t = 0$. Assume $A(\boldsymbol{s}, i, t') \leq F_g(\boldsymbol{s}, t' + 1)$ holds for any $0 \leq t' \leq t$, and any status $\boldsymbol{s}$. Our goal is to prove that $A(\boldsymbol{s}, i, t + 1) \leq A(\boldsymbol{s}, i^*, t + 1) = F_g(\boldsymbol{s}, t + 2)$. We expand $A(\boldsymbol{s}, i, t + 1)$ as follows.

$$
\begin{aligned}
A(\boldsymbol{s}, i, t + 1) &= (1 - s_i)F_g(\boldsymbol{s}, t + 1) + s_i F_g(\boldsymbol{s} - \mu_i \boldsymbol{I}_i, t + 1) + s_i \Delta_c \\
&= (1 - s_i)\left((1 - s_{i^*})F_g(\boldsymbol{s}, t) + s_{i^*} F_g(\boldsymbol{s} - \mu_{i^*} \boldsymbol{I}_{i^*}, t) + s_{i^*} \Delta_c\right) \\
&\quad + s_i((1 - s_{i^*})F_g(\boldsymbol{s} - \mu_i \boldsymbol{I}_i, t) + s_{i^*} F_g(\boldsymbol{s} - \mu_i \boldsymbol{I}_i - \mu_{i^*} \boldsymbol{I}_{i^*}, t) + s_{i^*} \Delta_{c+1}) \\
&\quad + s_i \Delta_c.
\end{aligned}
$$

Here $\Delta_{c+1} = f(c(\psi) + 2) - f(c(\psi) + 1)$. Above expansion of $A(i, t + 1)$ is based on Lemma 4. We expand $A(\boldsymbol{s}, i^*, t + 1)$ as follows.

$$
\begin{aligned}
A(\boldsymbol{s}, i^*, t+1) &= (1 - s_{i^*}) F_g(\boldsymbol{s}, t+1) + s_{i^*} F_g(\boldsymbol{s} - \mu_{i^*} \boldsymbol{I}_{i^*}, t+1) + s_{i^*} \Delta_c \\
&\geq (1 - s_{i^*}) \left( (1 - s_i) F_g(\boldsymbol{s}, t) + s_i F_g(\boldsymbol{s} - \mu_i \boldsymbol{I}_i, t) + s_i \Delta_c \right) \\
&\qquad\qquad\qquad\qquad\qquad\qquad\qquad\qquad \text{(assumption } A(\boldsymbol{s}, i, t) \leq F_g(\boldsymbol{s}, t+1)) \\
&\quad + s_{i^*} \left( (1 - s_i) F_g(\boldsymbol{s} - \mu_{i^*} \boldsymbol{I}_{i^*}, t) + s_i F_g(\boldsymbol{s} - \mu_{i^*} \boldsymbol{I}_{i^*} - \mu_i \boldsymbol{I}_i, t) + s_i \Delta_{c+1} \right) \\
&\qquad\qquad\qquad\qquad\quad \text{(assumption } A(\boldsymbol{s} - \mu_{i^*} \boldsymbol{I}_{i^*}, i, t) \leq F_g(\boldsymbol{s} - \mu_{i^*} \boldsymbol{I}_{i^*}, t+1)) \\
&\quad + s_{i^*} \Delta_c \\
&= A(i, t+1).
\end{aligned}
$$

This completes the proof. ∎

**Remarks.** During the rebuttal of this paper, we realized that Bubeck et al. [5] applied similar inductive reasoning techniques to prove the optimality of the greedy policy for their optimal discovery problem (Lemma 2 of [5]). To quantitatively measure how good is the greedy policy, we also give a formula to show the exact difference between $A(\boldsymbol{s}, i, t)$ and $A(\boldsymbol{s}, i^*, t)$ in Sec. B.3.

## B.2 Computation of expected reward

Lemma 4 indicates $r_{\pi^g}(\boldsymbol{\mu})$ can be computed in a recursive way. However, the recursive method has time complexity $O(2^K)$. It is impractical when $K$ is large. In the following we show that the expected reward of policy $\pi^g$ can be computed in polynomial time.

### B.2.1 Transition probability list of greedy policy

Assume we explore the communities via the greedy policy when the communities already have partial realization $\psi$. Define $s_{i,0} = 1 - \mu_i c_i(\psi)$ and $\boldsymbol{s}_0 = (s_{1,0}, \ldots, s_{m,0})$. The greedy policy will choose community $i_0^*$ to explore, where $i_0^* \in \arg\max_i s_{i,0}$. After one step exploration, the communities stay at the same status $\boldsymbol{s}_0$ with probability $q_0 := 1 - s_{i_0^*}$. The communities transit to next status $\boldsymbol{s}_1 := \boldsymbol{s}_0 - \mu_{i_0^*} \boldsymbol{I}_{i_0^*}$ with probability $p_0 := s_{i_0^*}$. We recursively define $\boldsymbol{s}_{t+1}$ as $\boldsymbol{s}_t - \mu_{i_t^*} \boldsymbol{I}_{i_t^*}$, where $i_t^* \in \arg\max_i s_{i,t}$. We call $p_t := \max_i s_{i,t}$ the *transition probability* and $q_t := 1 - p_t$ the *loop probability*. Each time the communities transit to next status, a new member will be met. During the exploration, the number of different statuses the communities can stay is at most $1 + \sum_i d_i - c_i(\psi)$ since there are $D := \sum_i d_i - c_i(\psi)$ unseen members in total. Based on above discussion, we define a *transition probability list* $\mathcal{P}(\pi^g, \psi) := (p_0, \ldots, p_D)$, where $p_D \equiv 0$. The list $\mathcal{P}(\pi^g, \psi)$ is unique for any initial partial realization $\psi$. Fig. 1 gives an example to demonstrate statuses and the list $\mathcal{P}(\pi^g, \psi)$.

**Corollary 1.** *Let $\psi$ be any partial realization corresponding to the status $\boldsymbol{s} = (s_1, \ldots, s_m)$. The number of unseen members $\sum_i d_i - c_i(\psi)$ is denoted as $D$. The probability list $\mathcal{P}(\pi^g, \psi) = (p_0, \ldots, p_D)$ can be obtained by sorting $\cup_{i=1}^m \{s_i, s_i - \mu_i, \ldots, \mu_i\} \cup \{0\}$ in descending order.*

Corollary 1 is an important observation based on the definition of transition probability list.

Figure 1: Illustration with $\boldsymbol{d} = (3, 4)$ and empty partial realization. The initial status is $(1, 1)$. The list $\mathcal{P}(\pi^g, \emptyset) = (1, 1, 3/4, 2/3, 1/2, 1/3, 1/4, 0)$.

### B.2.2 Compute the expected reward efficiently

**Lemma 6.** *Let $\psi$ be a partial realization and $\boldsymbol{s}_0$ be the corresponding status. The number of unseen members is denoted as $D = \sum_i d_i - c_i(\psi)$. The transition probability list is $\mathcal{P}(\pi^g, \psi) = (p_0, \ldots, p_D)$. Then*

$$
F_{\pi^g}(\psi, t) = F_g(\boldsymbol{s}_0, t) = \sum_{j=0}^{\min\{t, D\}} (f(j + c(\psi)) - f(c(\psi))) \times \left( \Pi_{l=0}^{j-1} p_j \right) \times \left( \sum_{I \in \mathcal{I}(j, t-j)} \Pi_{l \in I} q_l \right),
$$

*where $q_l = 1 - p_l$ and $\mathcal{I}(j, t-j)$ consists of subsets of multi-set $\{0, \ldots, j\}^{t-j}$ with fixed size $t - j$.*

**Proof.** When the communities ends at status $s_j$, we meet $j$ distinct members. Let $\Pr(s_j\square)$ be the probability for this event. We can the *transition step* as the communities transit to a new status, and the *loop step* as the communities stay at the same status. When the communities ends at status $s_j$, we have $j$ transition steps and $t - j$ loop steps. The communities takes loops at statuses $\{s_0, \ldots, s_j\}$. Hence,

$$\Pr(s_j\square) = \sum_{I \in \mathcal{I}(j, t-j)} \Pi_{l=0}^{j-1} p_j \cdot \Pi_{l \in I}(1 - p_l) = \Pi_{l=0}^{j-1} p_j \times \sum_{I \in \mathcal{I}(j, t-j)} \Pi_{l \in I} q_l.$$

The reward $F_{\pi^g}(\psi, t) = \sum_{j=1}^{\min\{t, D\}} (f(j + c(\psi)) - f(c(\psi))) \times \Pr(s_j\square)$. $\blacksquare$

For later analysis, we define the *loop probability*

$$L(\{q_0, \ldots, q_j\}, t) := \sum_{I \in \mathcal{I}(j, t)} \Pi_{l \in I} q_l$$

since $\sum_{I \in \mathcal{I}(j, t)} \Pi_{l \in I} q_l$ is just a function of $\{q_0, \ldots, q_j\}$ and $t$ ($t \geq 1$). Actually, $L(\{q_0, \ldots, q_j\}, t)$ aggregates the product of all possible $t$ elements in $\{q_0, \ldots, q_j\}$. Note that each element in $\{q_0, \ldots, q_j\}$ can be chosen multiple times. W.l.o.g, we define $L(\{q_0, \ldots, q_j\}, t) = 1$ and $\Pi_{l=0}^{t-1} p_l = 1$ when $t = 0$. Based on the definition, we can write $L(\{q_0, \ldots, q_j\}, t)$ in a recursive way as follows.

$$L(\{q_0, \ldots, q_j\}, t) = \sum_{s=0}^{t} q_a^s L(\{q_0, \ldots, q_j\} \backslash \{q_a\}, t - s). \tag{10}$$

Here $a \in \{0, \ldots, j\}$. According to Eq. 10, the probability $\sum_{I \in \mathcal{I}(j, t-j)} \Pi_{l \in I} q_l$ can be computed in $O((t - j)j^2)$ via *dynamic programming*. Hence $r_{\pi^g}(\boldsymbol{\mu}) = F_g((1, \ldots, 1), K)$ can be computed in $O(K \min\{K, D\}^2)$ according to Lemma 6.

## B.3   Reward gap between optimal policy and sub-optimal policy

Recall that $A(s, i, t)$ is the expected reward of the following adaptive process.

1. At the first step, choose an arbitrary community $C_i$ (different from $C_{i^*}$) to explore.

2. From the second step to the $(t + 1)$-th step, explore communities with the greedy policy $\pi^g$.

Here $s$ is the initial status of the communities. Lemma 5 only proves that $A(s, i, t) \leq F_g(s, t + 1)$. In the following, we aim to answer the following question:

- How much is $F_g(s, t + 1)$ larger than $A(s, i, t)$?

### B.3.1   Analysis of loop probability

The following two corollaries show the basic properties of the *loop probability*.

**Corollary 2.** *For a transition probability list $\mathcal{P}(\pi^g, \psi) = (p_0, \ldots, p_D)$, we have*

$$\sum_{j=0}^{M} p_0 \times \cdots \times p_{j-1} \times L(\{q_0, \ldots, q_j\}, t - j) = 1,$$

*where $q_j = 1 - p_j$ and $M = \min\{t, D\}$.*

Corollary 2 says the probabilities that the communities ends at status $\{s_0, \ldots, s_D\}$ sums up to 1.

**Corollary 3.** *For a transition probability list $\mathcal{P}(\pi^g, \psi) = (p_0, \ldots, p_D)$ and $a, b \in \{0, \ldots, j\}$ ($j \leq D, t \geq 1$), we have*

$$L(\{q_0, \ldots, q_j\} \backslash \{q_a\}, t) - L(\{q_0, \ldots, q_j\} \backslash \{q_b\}, t) = (q_b - q_a) L(\{q_0, \ldots, q_j\}, t - 1),$$

*where $D = \sum_i d_i - c_i(\psi)$ and $q_j = 1 - p_j$.*

**Proof.** We prove the corollary according to Eq. (10).

$$L(\{q_0,\ldots,q_j\}\backslash\{q_a\},t) - L(\{q_0,\ldots,q_j\}\backslash\{q_b\},t)$$

$$= \sum_{s=0}^{t}(q_b^s - q_a^s)L(\{q_0,\ldots,q_j\}\backslash\{q_a,q_b\},t-s) \qquad\qquad \text{(by Eq. (10))}$$

$$= \sum_{s=0}^{t-1}(q_b^{s+1} - q_a^{s+1})L(\{q_0,\ldots,q_j\}\backslash\{q_a,q_b\},t-s-1) \qquad\qquad \text{(replace } s-1 \text{ as } s')$$

$$=(q_b - q_a)\sum_{s=0}^{t-1}\sum_{m=0}^{s} q_b^{s-m}q_a^{m}L(\{q_0,\ldots,q_j\}\backslash\{q_a,q_b\},t-1-s) \quad \text{(sum of geometric sequence)}$$

$$=(q_b - q_a)L(\{q_0,\ldots,q_j\},t-1). \qquad\qquad \text{(by definition or expanding Eq. (10))}$$

This completes the proof. ∎

### B.3.2 Pseudo reward

**Lemma 7.** *For a transition probability list* $\mathcal{P}(\pi^g,\psi) = (p_0,\ldots,p_D)$ *and a non-decreasing function* $f(x)$*, a pseudo reward* $R(k)$ *is defined as*

$$R(k) = q_k\sum_{j=0}^{M} f(j) \times p_0 \times \cdots \times p_{j-1} \times L(\{q_0,\ldots,q_j\},t-j)$$

$$+ p_k\sum_{j=0}^{k-1} f(j+1) \times p_0 \times \cdots \times p_{j-1} \times L(\{q_0,\ldots,q_j\},t-j)$$

$$+ \sum_{j=k}^{M'} f(j+1) \times p_0 \times \cdots \times p_{j} \times L(\{q_0,\ldots,q_{j+1}\}\backslash\{q_k\},t-j),$$

*where* $M = \min\{D,t\}$ *and* $M' = \{D-1,t\}$*. We claim that for* $0 \le k \le M-1$*,*

$$R(k) - R(k+1) = (p_k - p_{k+1})\left(\sum_{j=0}^{k}(f(j+1)-f(j))p_0 \times \cdots \times p_{j-1} \times L(\{q_0,\ldots,q_j\},t-j)\right).$$

**Proof.** We expand $R(k) - R(k+1)$ as follows using the definition.

$$R(k) - R(k+1)$$

$$= -(p_k - p_{k+1})\sum_{j=0}^{M} f(j) \times p_0 \times \cdots \times p_{j-1} \times L(\{q_0,\ldots,q_j\},t-j)$$

$$+ (p_k - p_{k+1})\sum_{j=0}^{k-1} f(j+1) \times p_0 \times \cdots \times p_{j-1} \times L(\{q_0,\ldots,q_j\},t-j)$$

$$+ f(k+1) \times p_0 \times \cdots \times p_{k-1} \times p_k \times L(\{q_0,\ldots,q_{k+1}\}\backslash\{q_k\},t-k) \qquad \text{(from } R(k))$$

$$- f(k+1) \times p_0 \times \cdots \times p_{k-1} \times p_{k+1} \times L(\{q_0,\ldots,q_k\},t-k) \qquad \text{(from } R(k+1))$$

$$+ \underbrace{\sum_{j=k+1}^{M'} f(j+1) \times p_0 \times \cdots \times p_{j} \times (L(\{q_0,\ldots,q_{j+1}\}\backslash\{q_k\},t-j)}_{(p_k-p_{k+1})\sum_{j=k+1}^{M-1} f(j+1)\times p_0 \times\cdots\times p_j \times L(\{q_0,\ldots,q_{j+1}\},t-j-1)}$$

$$\qquad\qquad\qquad -L(\{q_0,\ldots,q_{j+1}\}\backslash\{q_{k+1}\},t-j)).$$

The last line of above equation can be rewritten with the Corollary 3. The summation from $j = k+1$ to $j = M-1$ in the last line cancels out with the second line when $j = k+2$ to $j = M$. The

summation from $j = 0$ to $j = k$ in the second line can be combined with the third line. We continue the computation of $R(k) - R(k + 1)$ by rearranging its expansion.

$$R(k) - R(k + 1)$$
$$= - (p_k - p_{k+1}) f(k + 1) \times p_0 \times \cdots \times p_k \times L(\{q_0, \ldots, q_{k+1}\}, t - k - 1)$$
$$- (p_k - p_{k+1}) f(k + 1) \times p_0 \times \cdots \times p_{k-1} \times L(\{q_0, \ldots, q_k\}, t - k)$$
$$+ (p_k - p_{k+1}) \sum_{j=0}^{k} (f(j + 1) - f(j)) \times p_0 \times \cdots \times p_{j-1} \times L(\{q_0, \ldots, q_j\}, t - j)$$
$$+ f(k + 1) \times p_0 \times \cdots \times p_{k-1} \times p_k \times L(\{q_0, \ldots, q_{k+1}\} \backslash \{q_k\}, t - k)$$
$$- f(k + 1) \times p_0 \times \cdots \times p_{k-1} \times p_{k+1} \times L(\{q_0, \ldots, q_k\}, t - k).$$

Define $\Delta_k$ as the sum of the 2nd, 3rd, 5th, 6th line in above equation. We have

$$R(k) - R(k + 1)$$
$$= \left. \begin{array}{l} -(p_k - p_{k+1}) \times f(k + 1) \times p_0 \times \cdots \times p_{k-1} \times L(\{q_0, \ldots, q_k\}, t - k) \\ -(p_k - p_{k+1}) \times f(k + 1) \times p_0 \times \cdots \times p_k \times L(\{q_0, \ldots, q_{k+1}\}, t - k - 1) \\ +f(k + 1) \times p_0 \times \cdots \times p_{k-1} \times p_k \times L(\{q_0, \ldots, q_{k+1}\} \backslash \{q_k\}, t - k) \\ -f(k + 1) \times p_0 \times \cdots \times p_{k-1} \times p_{k+1} \times L(\{q_0, \ldots, q_k\}, t - k) \end{array} \right\} \triangleq \Delta_k$$
$$+ (p_k - p_{k+1}) \sum_{j=0}^{k} (f(j + 1) - f(j)) \times p_0 \times \cdots \times p_{j-1} \times L(\{q_0, \ldots, q_j\}, t - j).$$

We rewrite $\Delta_k$ as follows.

$$\Delta_k / f(k + 1) = p_0 \times \cdots \times p_{k-1} \times p_k \times L(\{q_0, \ldots, q_{k+1}\} \backslash \{q_k\}, t - k)$$
$$\left. \begin{array}{l} -p_0 \times \cdots \times p_{k-1} \times p_k \times L(\{q_0, \ldots, q_k\}, t - k) \\ +p_0 \times \cdots \times p_{k-1} \times p_k \times L(\{q_0, \ldots, q_k\}, t - k) \end{array} \right\} \text{ cancel each other}$$
$$- (p_k - p_{k+1}) \times p_0 \times \cdots \times p_{k-1} \times L(\{q_0, \ldots, q_k\}, t - k)$$
$$- (p_k - p_{k+1}) \times p_0 \times \cdots \times p_k \times L(\{q_0, \ldots, q_{k+1}\}, t - k - 1)$$
$$- p_0 \times \cdots \times p_{k-1} \times p_{k+1} \times L(\{q_0, \ldots, q_k\}, t - k)).$$

According to Corollary 3, the first line and the second line of above equation equals to $(p_k - p_{k+1}) \times p_0 \times \cdots \times p_k \times L(\{q_0, \ldots, q_{k+1}\}, t - k - 1)$, which cancels out with the fifth line. Hence, we have

$$\Delta_k / f(k + 1) = p_0 \times \cdots \times p_{k-1} \times p_k \times L(\{q_0, \ldots, q_k\}, t - k)$$
$$- (p_k - p_{k+1}) \times p_0 \times \cdots \times p_{k-1} \times L(\{q_0, \ldots, q_k\}, t - k)$$
$$- p_0 \times \cdots \times p_{k-1} \times p_{k+1} \times L(\{q_0, \ldots, q_k\}, t - k))$$
$$= 0.$$

With above result of $\Delta_k = 0$, we prove that

$$R(k) - R(k + 1)$$
$$= (p_k - p_{k+1}) \left( \sum_{j=0}^{k} (f(j + 1) - f(j)) p_0 \times \cdots \times p_{j-1} \times L(\{q_0, \ldots, q_j\}, t - j) \right) \geq 0.$$

This completes the proof. ∎

### B.3.3 Reward gap

Let $\psi, \psi', \psi''$ be any partial realization corresponding to the status $s, s - \mu_{i^*} \boldsymbol{I}_{i^*}, s - \mu_i \boldsymbol{I}_i$ respectively. Define $\mathcal{P}(\pi^g, \psi) = (p_0, \ldots, p_D)$, where $D = \sum_i d_i - c_i(\psi)$. Recalling Corollary 1, we know that $(p_0, \ldots, p_{D-1})$ can be obtained by sorting $\cup_{i=1}^{m} \{s_i, s_i - \mu_i, \ldots, \mu_i\}$. Assume the first time $s_i$ appear in $(p_0, \ldots, p_D)$ is the $k$-th entry, i.e., $k = \min\{k' : 0 \leq k' \leq D, p_{k'} = s_i\}$. According to Corollary 1, we have the following.

$$\mathcal{P}(\pi^g, \psi') = (p_1, \ldots, p_D),$$
$$\mathcal{P}(\pi^g, \psi'') = (p_0, \ldots, p_{k-1}, p_{k+1}, \ldots, p_D).$$

Note that $p_0 = s_{i^*}$ and $p_k = s_i$. Let $M = \min\{D, t\}$, $M' = \min\{D-1, t\}$, and $f'(j) = f(j + c(\psi)) - f(c(\psi))$. The second line of Eq. (9) is

$$R_1 = q_0 \sum_{j=0}^{M} f'(j) \times p_0 \times \cdots \times p_{j-1} \times L(\{q_0, \ldots, q_j\}, t - j) \qquad ((1 - s_{i^*})F_g(\boldsymbol{s}, t))$$

$$+ p_0 \sum_{j=0}^{M'} f'(j+1) \times p_1 \times \cdots \times p_j \times L(\{q_1, \ldots, q_{j+1}\}, t - j). \quad (s_{i^*} F_g(\boldsymbol{s} - \mu_{i^*} \boldsymbol{I}_{i^*}, t) + s_{i^*} \Delta_c)$$

In fact, $R_1 = F_g(\boldsymbol{s}, t+1)$ based on Lemma 4. The first line of Eq. (9) is

$$R_2 = q_k \sum_{j=0}^{M} f'(j) \times p_0 \times \cdots \times p_{j-1} \times L(\{q_0, \ldots, q_j\}, t - j)$$

$$+ p_k \sum_{j=0}^{k-1} f'(j+1) \times p_0 \times \cdots \times p_{j-1} \times L(\{q_0, \ldots, q_j\}, t - j)$$

$$+ \sum_{j=k}^{M'} f'(j+1) \times p_0 \times \cdots \times p_j \times L(\{q_0, \ldots, q_{j+1}\} \backslash \{q_k\}, t - j).$$

Our goal is to measure the gap $R_1 - R_2$. Let $\text{Prob}_{\boldsymbol{s},t}(i)$ be the probability we can meet $i$ distinct members if we explore communities (whose initial status is $\boldsymbol{s}$) with greedy policy for $t$ steps. According to Lemma 7, we have

$$F_g(\boldsymbol{s}, t+1) - A(\boldsymbol{s}, i, t) = \sum_{j=0}^{k-1} (R(j) - R(j+1))$$

$$= \sum_{j=0}^{k-1} (p_j - p_{j+1}) \left( \sum_{o=0}^{j} (f'(o+1) - f'(o)) \text{Prob}_{\boldsymbol{s},t}(o) \right)$$

$$= \sum_{o=0}^{k-1} (f'(o+1) - f'(o)) \text{Prob}_{\boldsymbol{s},t}(o) \left( \sum_{j=o}^{k-1} p_j - p_{j+1} \right)$$

$$= \sum_{j=0}^{k-1} (f'(j+1) - f'(j))(p_j - p_k) \text{Prob}_{\boldsymbol{s},t}(j)$$

When the reward equals to the number of distinct members, we have

$$F_g(\boldsymbol{s}, t+1) - A(\boldsymbol{s}, i, t) = \sum_{j=0}^{k-1} (p_j - p_k) \text{Prob}_{\boldsymbol{s},t}(j).$$

Besides, the gap $F_g(\boldsymbol{s}, t+1) - A(\boldsymbol{s}, i, t)$ increases as $k$ increases, which means the worse choice we have at first, the larger reward gap we have at end.

## C  Basics of online learning problems

### C.1  Set size estimation by collision counting

Suppose we have a set $C_i = \{u_1, \cdots, u_{d_i}\}$ whose population $d_i$ is unknown. Let $u, v$ be two elements selected with replacement from $C_i$, and $Y_{u,v}$ denote a random variable that takes value 1 if $u = v$ (a collision) and 0 otherwise. The expectation of $Y_{u,v}$ equals to $\frac{1}{d_i}$, i.e., $\mathbb{E}[Y_{u,v}] = \frac{1}{d_i}$. Assume we sample $k_i$ elements with replacement uniformly at random from set $C_i$. Let $\mathcal{S}_i$ be the set of samples. With the sample $\mathcal{S}_i$, we compute the estimator for $d_i$ as

$$\hat{d}_i = \frac{k_i(k_i - 1)}{2X_i},$$

here $X_i = \sum_{u \in \mathcal{S}_i, v \in \mathcal{S}_i \setminus \{u\}} Y_{u,v}$ is the number of collisions in $\mathcal{S}_i$. According to the Jensen's inequality[3], we have $d_i \leq \mathbb{E}[\hat{d}_i]$, i.e., $\hat{d}_i$ is a biased estimator. The estimator is invalid when $X_i = 0$. Since the equality only occurs when $\text{Var}[X_i] = 0$, which is not the case Here. We have $d_i < \mathbb{E}[\hat{d}_i]$.

**Independence.** Let $\mathcal{S}_i = \{v_1, \cdots, v_{k_i}\}$. For the two random variable $Y_{v_x,v_y}$ ($1 \leq x < y \leq k_i$) and $Y_{v_{x'},v_{y'}}$ ($1 \leq x' < y' \leq k_i$), we consider three difference cases.

1. There are $\binom{k_i}{2}$ occurrences when $x = x', y = y'$. Here $\mathbb{E}[Y_{v_x,v_y} Y_{v_{x'},v_{y'}}] = 1/d_i$.

2. There are $6\binom{k_i}{3}$ occurrences when $x = x', y \neq y'$ or $x \neq x', y = y'$. $\mathbb{E}[Y_{v_x,v_y} Y_{v_{x'},v_{y'}}] = 1/d_i^2$.

3. There are $6\binom{k_i}{4}$ occurrences when $x \neq x', y \neq y'$. Here $\mathbb{E}[Y_{v_x,v_y} Y_{v_{x'},v_{y'}}] = 1/d_i^2$.

We say that pairs $(v_x, v_y)$ and $(v_{x'}, v_{y'})$ are different if $x \neq x'$ or $y \neq y'$. When $(v_x, v_y)$ and $(v_{x'}, v_{y'})$ are different, we have $\mathbb{E}[Y_{v_x,v_y} Y_{v_{x'},v_{y'}}] = \mathbb{E}[Y_{v_x,v_y}]\mathbb{E}[Y_{v_{x'},v_{y'}}] = 1/d_i^2$. Above discussion indicates that the $\binom{k_i}{2}$ pairs of random variables obtained from $\mathcal{S}_i$ are 2-*wise independent*.

**Variance.** We compute the variance $\text{Var}[X_i] = \mathbb{E}[X_i^2] - \mathbb{E}^2[X_i]$ in the following.

$$\text{Var}[X_i] = \frac{k_i(k_i-1)}{2d_i} + \frac{k_i(k_i-1)(k_i-2)}{d_i^2} + \frac{k_i(k_i-1)(k_i-2)(k_i-3)}{4d_i^2} - \frac{k_i^2(k_i-1)^2}{4d_i^2}$$

$$= \binom{k_i}{2}\frac{1}{d_i}(1 - \frac{1}{d_i}) = \binom{k_i}{2}\text{Var}[Y_{u,v}].$$

**Collision** Since the estimator is based on the collision counting, we need to ensure that $X_i > 0$ with high probability. Let $B_{k_i}$ denote the event that the $k_i$ samples $\{v_1, \ldots, v_{k_i}\}$ are distinct. We have

$$\text{Pr}\{B_k\} = 1 \cdot (1 - \frac{1}{d_i})(1 - \frac{2}{d_i}) \cdots (1 - \frac{k_i-1}{d_i}) \leq e^{-1/d_i} e^{-2/d_i} \cdots e^{-(k_i-1)d_i}$$

$$= e^{-\sum_{j=1}^{k_i-1} j/d_i} = e^{-k_i(k_i-1)/2d_i}.$$

To ensure that $X_i > 0$ with probability no less than $1 - \delta$, we have

$$k_i \geq \left(1 + \sqrt{8d_i \ln\frac{1}{\delta} + 1}\right)/2.$$

### C.2 Concentration bound for variables with local dependence

Note that the pairs $Y_{u,v}$ and $Y_{u',v'}$ are not mutually independent. Actually, their dependence can be described with a *dependence graph* [9, 15]. The Chernoff-Hoeffding bound in [14] can not be used directly for our estimator of $\mu_i$. In the following, we present a concentration bound that is applicable to our problem.

**Definition 1** (U-statistics). *Let $\xi_1, \ldots, \xi_n$ be independent random variables, and let*

$$X := \sum_{1 \leq i_1 \leq \cdots \leq i_d} f_{i_1, \ldots, i_d}(\xi_{i_1}, \ldots, \xi_{i_d}).$$

**Lemma 8** (Chapter 3.2 [9]). *If $a \leq f_{i_1, \ldots, i_d}(\xi_{i_1}, \ldots, \xi_{i_d}) \leq b$ for every $i_1, \ldots, i_d$ for some reals $a \leq b$, we have*

$$\text{Pr}\left\{|X - \mathbb{E}[X]| \geq \epsilon\binom{n}{d}\right\} \leq 2\exp\left(\frac{-2\lfloor n/d \rfloor \epsilon^2}{(b-a)^2}\right).$$

In our problem, if we get $k_i$ samples from set $C_i$, then the number of collisions satisfies

$$\text{Pr}\left\{|X_i - \mathbb{E}[X_i]| \geq \epsilon\binom{k_i}{2}\right\} \leq 2\exp\left(-2\lfloor k_i/2 \rfloor \epsilon^2\right).$$

Above inequality indicates that the actual number of independent pairs is $\lfloor k_i/2 \rfloor$ when using collisions in $k_i$ samples to estimate $\mu_i$.

# D  Regret Analysis for Non-Adaptive Problem

## D.1  Supporting Corollaries

**Corollary 4.** *For action $\boldsymbol{k}$ with $\sum_{i=1}^{m} k_i = K$ and $k_i \geq 1$, we have $\sum_{i=1}^{m} \binom{k_i}{2} \leq \binom{K-m+1}{2}$.*

**Proof.** We prove the corollary by simple calculation.

$$
\begin{aligned}
\sum_{i=1}^{m} \binom{k_i}{2} - \binom{K-m+1}{2} &= \frac{1}{2}\left( \sum_{i=1}^{m} k_i(k_i-1) - \left(1 + \sum_{i=1}^{m}(k_i-1)\right)\left(\sum_{i=1}^{m}(k_i-1)\right) \right) \\
&= \frac{1}{2}\left( \sum_{i=1}^{m}(k_i-1)^2 - \left(\sum_{i=1}^{m}(k_i-1)\right)^2 \right) \leq 0. \qquad \blacksquare
\end{aligned}
$$

## D.2  Basics

To compare with the CUCB algorithm introduced in [20] for general CMAB problem, we propose an revised Algo. 3 that is consistent with the CUCB algorithm in [20]. We revise the Line 6-8 in Algo. 3 as follows.

Line 6:   For $i \in [m], T_i \leftarrow T_i + \mathbb{1}\{|\mathcal{S}_i| > 1\}$,

Line 7:   For $i \in [m]$ and $|\mathcal{S}_i| > 1, X_{i,t} \leftarrow \sum_{x=1}^{\lfloor |\mathcal{S}_i| \rfloor/2} \mathbb{1}\{u_{2x-1} = u_{2x}\}/\lfloor |\mathcal{S}_i/2| \rfloor$,     (11)

Line 8: For $i \in [m]$ and $|\mathcal{S}_i| > 1, \hat{\mu}_i \leftarrow \hat{\mu}_i + (X_{i,t} - \hat{\mu}_i)/T_i$.

Note that $\hat{\mu}_i$ in Eq. (11) is also an unbiased estimator of $\mu_i$. Then we can obtain the regret bound of the revised Algo. 3 by applying the Theorem 4 in the extended version of [20] directly.

$$
\text{Reg}_{\boldsymbol{\mu}}(T) \leq \sum_{i=1}^{m} \frac{48\binom{K-m+1}{2}^2 m \ln T}{\Delta_{\min}^i} + 2\binom{K-m+1}{2}m + \frac{\pi^2}{3} \cdot m \cdot \Delta_{\max}.
$$

We add superscript $r$ to differentiate the corresponding random variables in the revised Algo. 3 from the original ones. E.g., $T_{i,t}^r$ is the value of $T_i^r$ in the revised Algo. 3 at the end of round $t$. Recall that $K' = K - m + 1$, which is the maximum exploration times for a community in each round.

## D.3  Proof framework

We first introduce a definition which describes the event that $\hat{\mu}_{i,t-1}$ ($\hat{\mu}_{i,t-1}^r$) is accurate at the beginning of round $t$.

**Definition 2.** *We say that the sampling is nice at the beginning of round $t$ if for every community $i \in [m]$, $|\hat{\mu}_{i,t-1} - \mu_i| \leq \rho_{i,t}$ (resp. $|\hat{\mu}_{i,t-1}^r - \mu_i| \leq \rho_{i,t}^r$), where $\rho_{i,t} = 2\sqrt{\frac{3\ln t}{2T_{i,t-1}}}$ (resp. $\rho_{i,t}^r = 2\sqrt{\frac{3\ln t}{2T_{i,t-1}^r}}$) in round $t$. Let $\mathcal{N}_t$ (resp. $\mathcal{N}_t^r$) be such event.*

**Lemma 9.** *For each round $t \geq 1$, $\Pr\{\neg\mathcal{N}_t\} \leq 2m\lfloor K'/2 \rfloor t^{-2}$ (resp. $\Pr\{\neg\mathcal{N}_t^r\} \leq 2mt^{-2}$).*

**Proof.** For each round $t \geq 1$, we have

$$\Pr\{\neg \mathcal{N}_t\} = \Pr\left\{\exists i \in [m], |\hat{\mu}_{i,t-1} - \mu_i| \geq \sqrt{\frac{3 \ln t}{2T_{i,t-1}}}\right\}$$

$$\leq \sum_{i \in [m]} \Pr\left\{|\hat{\mu}_{i,t-1} - \mu_i| \geq \sqrt{\frac{3 \ln t}{2K_{i,t-1}}}\right\}$$

$$= \sum_{i \in [m]} \sum_{k=1}^{(t-1)\lfloor K'/2 \rfloor} \Pr\left\{T_{i,t-1} = k, |\hat{\mu}_{i,t-1} - \mu_i| \geq \sqrt{\frac{3 \ln t}{2T_{i,t-1}}}\right\}$$

$$\leq \sum_{i \in [m]} \sum_{k=1}^{(t-1)\lfloor K'/2 \rfloor} \frac{2}{t^3} < 2m\lfloor K'/2 \rfloor t^{-2}. \qquad \text{(Hoeffding's inequality [14])}$$

When $T_{i,t-1} = k$, $\hat{\mu}_{i,t}$ is the average of $k$ i.i.d. random variables $Y_i^{[1]}, \cdots, Y_i^{[k]}$, where $Y_i^{[j]}$ is a random variable that indicates whether two members selected with replacement from $C_i$ are the same. Since each community is explored at most $K'$ times in each round, $T_{i,t-1} \leq (t-1)\lfloor K'/2 \rfloor$. The last line leverages the Hoeffding's inequality [14]. By replacing the summation range $k \in [1, (t-1)\lfloor K'/2 \rfloor]$ with $k \in [1, (t-1)]$ in the 3rd line of above equation, we have $\Pr\{\neg \mathcal{N}_t^T\} \leq 2mt^{-2}$. ∎

Secondly, we use the monotonicity and bounded smoothness properties to bound the reward gap $\Delta_{\boldsymbol{k}_t} = r_{\boldsymbol{k}^*}(\boldsymbol{\mu}) - r_{\boldsymbol{k}_t}(\boldsymbol{\mu})$ between our action $\boldsymbol{k}_t$ and the optimal action $\boldsymbol{k}^*$.

**Lemma 10.** *If the event $\mathcal{N}_t$ holds in round t, we have*

$$\Delta_{\boldsymbol{k}_t} \leq \sum_{i=1}^{m} \binom{k_{i,t}}{2} \kappa_T(\Delta_{\min}^i, T_{i,t-1}).$$

*Here the function $\kappa_T(M, s)$ is defined as*

$$\kappa_T(M, s) = \begin{cases} 2 & \text{if } s = 0, \\ 2\sqrt{\frac{6 \ln t}{s}} & \text{if } 1 \leq s \leq l_T(M), \\ 0 & \text{if } s \geq l_T(M) + 1, \end{cases}$$

*where*

$$l_T(M) = \frac{24\binom{K'}{2}^2 \ln T}{M^2}.$$

**Proof.** By $\mathcal{N}_t$ (i.e., $\underline{\boldsymbol{\mu}}_t \leq \boldsymbol{\mu}$) and the monotonicity of $r_{\boldsymbol{k}}(\boldsymbol{\mu})$, we have

$$r_{\boldsymbol{k}_t}(\underline{\boldsymbol{\mu}}_t) \geq r_{\boldsymbol{k}^*}(\underline{\boldsymbol{\mu}}_t) \geq r_{\boldsymbol{k}^*}(\boldsymbol{\mu}) = r_{\boldsymbol{k}_t}(\boldsymbol{\mu}) + \Delta_{\boldsymbol{k}_t}.$$

Then by the *bounded smoothness* properties of reward function, we have

$$\Delta_{\boldsymbol{k}_t} \leq r_{\boldsymbol{k}_t}(\underline{\boldsymbol{\mu}}_t) - r_{\boldsymbol{k}_t}(\boldsymbol{\mu}) \leq \sum_{i=1}^{m} \binom{k_{i,t}}{2}(\mu_i - \underline{\mu}_{i,t}).$$

We intend to bound $\Delta_{\boldsymbol{k}_t}$ by bounding $\mu_i - \underline{\mu}_{i,t}$. Before doing so, we perform a transformation. Let $M_{\boldsymbol{k}_t} = \max_{i \in [m], k_{i,t} > 1} \Delta_{\min}^i$. Since the action $\boldsymbol{k}_t$ always satisfies $\Delta_{\boldsymbol{k}_t} \geq \max_{i \in [m], k_{i,t} > 1} \Delta_{\min}^i$,

Figure 2: Demonstration of the regret summation $\sum_{t=2}^{T} \lfloor k_{i,t}/2 \rfloor \kappa_T(\Delta_{\min}^i, T_{i,t-1})$. It is obvious that when $k_{i,t} = K'$, then the shaded area (colored with orange) covered by the rectangles is maximized.

we have $\Delta_{\boldsymbol{k}_t} \geq M_{\boldsymbol{k}_t}$. So $\sum_i \binom{k_{i,t}}{2}(\mu_i - \underline{\mu}_{i,t}) \geq \Delta_{\boldsymbol{k}_t} \geq M_{\boldsymbol{k}_t}$. Therefore,

$$
\begin{aligned}
\Delta_{\boldsymbol{k}_t} &\leq \sum_{i=1}^{m} \binom{k_{i,t}}{2}(\mu_i - \underline{\mu}_{i,t}) \leq -M_{\boldsymbol{k}_t} + 2\sum_{i=1}^{m} \binom{k_{i,t}}{2}(\mu_i - \underline{\mu}_{i,t}) \\
&\leq -\frac{\sum_{i=1}^{m} \binom{k_{i,t}}{2}}{\binom{K'}{2}} M_{\boldsymbol{k}_t} + 2\sum_{i=1}^{m} \binom{k_{i,t}}{2}(\mu_i - \underline{\mu}_{i,t}) \qquad \text{(Corollary 4: } \sum_{i=1}^{m} \binom{k_{i,t}}{2} \leq \binom{K'}{2}) \\
&= 2\sum_{i=1}^{m} \binom{k_{i,t}}{2}\left[(\mu_i - \underline{\mu}_{i,t}) - \frac{M_{\boldsymbol{k}_t}}{K'(K'-1)}\right] \\
&\leq 2\sum_i \binom{k_{i,t}}{2}\left[(\mu_i - \underline{\mu}_{i,t}) - \frac{\Delta_{\min}^i}{K'(K'-1)}\right]. \qquad \text{(by definition of } M_{\boldsymbol{k}_t})
\end{aligned}
$$

By $\mathcal{N}_t$, we have $\mu_i - \underline{\mu}_{i,t} \leq \min\{2\rho_{i,t}, 1\}$. So

$$
\mu_i - \underline{\mu}_{i,t} - \frac{\Delta_{\min}^i}{K'(K'-1)} \leq \min\{2\rho_{i,t}, 1\} - \frac{\Delta_{\min}^i}{K'(K'-1)} \leq \min\left\{\sqrt{\frac{6\ln t}{T_{i,t-1}}}, 1\right\} - \frac{\Delta_{\min}^i}{K'(K'-1)}.
$$

If $T_{i,t-1} \leq l_T(\Delta_{\min}^i)$, we have $\mu_i - \underline{\mu}_{i,t} - \frac{\Delta_{\min}^i}{K'(K'-1)} \leq \min\left\{\sqrt{\frac{6\ln t}{T_{i,t-1}}}, 1\right\} \leq \frac{1}{2}\kappa_T(\Delta_{\min}^i, T_{i,t-1})$. If $T_{i,t-1} > l_T(\Delta_{\min}^i) + 1$, then $\sqrt{\frac{6\ln t}{T_{i,t-1}}} \leq \frac{\Delta_{\min}^i}{K'(K'-1)}$, so $(\mu_i - \underline{\mu}_{i,t}) - \frac{\Delta_{\min}^i}{K'(K'-1)} \leq 0 = \kappa_T(\Delta_{\min}^i, T_{i,t-1})$. In conclusion, we have

$$
\Delta_{\boldsymbol{k}_t} \leq \sum_{i=1}^{m} \binom{k_{i,t}}{2}\kappa_T(\Delta_{\min}^i, T_{i,t-1}). \qquad\blacksquare
$$

Above result is also valid for the revised Algo. 3, i.e., $\Delta_{\boldsymbol{k}_t}^r \leq \sum_{i=1}^{m} \binom{k_{i,t}}{2}\kappa_T(\Delta_{\min}^i, T_{i,t-1}^r)$. Our third step is to prove that when $\mathcal{N}_t$ (resp. $\mathcal{N}_t^r$) holds, the regret is bounded in $O(\ln T)$.

**Theorem 3.** *Algo. 3 with non-adaptive exploration method has regret as follows.*

$$
Reg_{\boldsymbol{\mu}}(T) \leq \sum_{i=1}^{m} \frac{48\binom{K'}{2}K\ln T}{\Delta_{\min}^i} + 2\binom{K'}{2}m + \frac{\lfloor \frac{K'}{2} \rfloor \pi^2}{3}m\Delta_{\max} = O\left(\sum_{i=1}^{m} \frac{K'^3 \log T}{\Delta_{\min}^i}\right). \quad (6)
$$

**Proof.** We first prove the regret when the event $\mathcal{N}_t$ holds. In each run, we have

$$\sum_{t=1}^{T} \mathbf{1}(\{\Delta_{\boldsymbol{k}_t} \wedge \mathcal{N}_t\}) \cdot \Delta_{\boldsymbol{k}_t} \leq \sum_{t=1}^{T} \sum_{i=1}^{m} \binom{k_{i,t}}{2} \kappa_T(\Delta_{\min}^i, T_{i,t-1})$$

$$= \sum_{i=1}^{m} \sum_{t' \in \{t|1 \leq t \leq T, k_{i,t} > 1\}} \binom{k_{i,t'}}{2} \kappa_T(\Delta_{\min}^i, T_{i,t'-1}).$$

Hence, we just assume $k_{i,t} > 1$ for $t > 0$.

$$\sum_{t=1}^{T} \mathbf{1}(\{\Delta_{\boldsymbol{k}_t} \wedge \mathcal{N}_t\}) \cdot \Delta_{\boldsymbol{k}_t} \leq \sum_{i=1}^{m} \sum_{t=1}^{T} \binom{k_{i,t}}{2} \kappa_T(\Delta_{\min}^i, T_{i,t-1})$$

$$\leq \sum_{i=1}^{m} 2 \binom{k_{i,1}}{2} + K' \sum_{i=1}^{m} \sum_{t=2}^{T} \frac{(k_{i,t}-1)}{2} \kappa_T(\Delta_{\min}^i, T_{i,t-1})$$

$$\leq 2m \binom{K'}{2} + K' \sum_{i=1}^{m} \sum_{t=2}^{T} \left\lfloor \frac{k_{i,t}}{2} \right\rfloor \kappa_T(\Delta_{\min}^i, T_{i,t-1}). \qquad \text{(Fig. 2)}$$

To maximize the summation $\sum_{t=2}^{T} \lfloor \frac{k_{i,t}}{2} \rfloor \kappa_T(\Delta_{\min}^i, T_{i,t-1})$, we just need to let $k_{i,t} = K'$ when $t > 1$.

$$\sum_{t=1}^{T} \mathbf{1}(\{\Delta_{\boldsymbol{k}_t} \wedge \mathcal{N}_t\}) \cdot \Delta_{\boldsymbol{k}_t} \leq 2m \binom{K'}{2} + K' \sum_{d=0}^{l_T(\Delta_{\min}^i)/\lfloor K'/2 \rfloor} \left\lfloor \frac{K'}{2} \right\rfloor \kappa_T\left(\Delta_{\min}^i, 1 + d\lfloor K'/2 \rfloor\right)$$

$$\leq 2m \binom{K'}{2} + K' \sum_{i=1}^{m} \sum_{d=0}^{l_{T,K}} \frac{\sqrt{24 \ln T} \lfloor K'/2 \rfloor}{\sqrt{1 + d\lfloor K'/2 \rfloor}} \qquad (l_{T,K} := \frac{l_T(\Delta_{\min}^i)}{\lfloor K'/2 \rfloor})$$

$$\leq 2m \binom{K'}{2} + K' \sum_{i=1}^{m} \int_{x=0}^{l_{T,K}} \frac{\sqrt{24 \lfloor K'/2 \rfloor \ln T}}{\sqrt{x}} dx$$

$$= 2m \binom{K'}{2} + K' \sum_{i=1}^{m} \sqrt{96 l_T(\Delta_{\min}^i, T) \ln T}$$

$$= 2m \binom{K'}{2} + \sum_{i=1}^{m} \frac{48 \binom{K'}{2} K' \ln T}{\Delta_{\min}^i}.$$

On the other hand, when $\mathcal{N}_t$ does not hold, we can bound the regret as $\Delta_{\max}$. Hence,

$$\mathbb{E}\left[\sum_{t=1}^{T} \mathbf{1}(\{\Delta_{\boldsymbol{k}_t} \wedge \neg\mathcal{N}_t\}) \cdot \Delta_{\boldsymbol{k}_t}\right] \leq \Delta_{\max} \sum_{t=1}^{T} 2m \lfloor K'/2 \rfloor t^{-2} \leq \frac{m \lfloor K'/2 \rfloor \pi^2}{3} \Delta_{\max}.$$

Based on above discussion, we have

$$\text{Reg}_{\boldsymbol{\mu}}(T) \leq \frac{m \lfloor K'/2 \rfloor \pi^2}{3} \Delta_{\max} + 2m \binom{K'}{2} + \sum_{i=1}^{m} \frac{48 \binom{K'}{2} K' \ln T}{\Delta_{\min}^i}. \qquad \blacksquare$$

**Theorem 7.** *The revised Algo. 3 has regret as follows.*

$$Reg_{\boldsymbol{\mu}}^r(T) \leq \sum_{i=1}^{m} \frac{48 \binom{K'}{2}^2 \ln T}{\Delta_{\min}^i} + 2\binom{K'}{2} m + \frac{\pi^2}{3} \cdot m \cdot \Delta_{\max}. \qquad (12)$$

**Proof.** We prove the regret when the event $\mathcal{N}_t^r$ holds. In each run, we have

$$\sum_{t=1}^{T} \mathbf{1}(\{\Delta_{\boldsymbol{k}_t}^r \wedge \mathcal{N}_t^r\}) \cdot \Delta_{\boldsymbol{k}_t}^r \leq \sum_{t=1}^{T} \sum_{i=1}^{m} \binom{k_{i,t}}{2} \kappa_T(\Delta_{\min}^i, T_{i,t-1}^r)$$

$$= \sum_{i=1}^{m} \sum_{s=0}^{T_{i,T}^r} \binom{k_{i,s}}{2} \kappa_T(\Delta_{\min}^i, s)$$

$$\leq 2m \binom{K'}{2} + \binom{K'}{2} \sum_{i=1}^{m} \sum_{s=1}^{l_T(\Delta_{\min}^i)} \sqrt{\frac{24 \ln T}{s}}$$

$$\leq 2m \binom{K'}{2} + \sum_{i=1}^{m} \frac{48 \binom{K'}{2}^2 \ln T}{\Delta_{\min}^i}.$$

On the other hand, $\Pr\{\neg \mathcal{N}_t^r\} \leq 2mt^{-2}$. Hence we have

$$\text{Reg}_{\boldsymbol{\mu}}^r(T) = \mathbb{E}\left[\sum_{t=1}^{T} \mathbf{1}(\{\Delta_{\boldsymbol{k}_t} \wedge \neg \mathcal{N}_t\}) \cdot \Delta_{\boldsymbol{k}_t}\right] + \mathbb{E}\left[\sum_{t=1}^{T} \mathbf{1}(\{\Delta_{\boldsymbol{k}_t} \wedge \mathcal{N}_t\}) \cdot \Delta_{\boldsymbol{k}_t}\right]$$

$$\leq \frac{m\pi^2}{3} \Delta_{\max} + 2m \binom{K'}{2} + \sum_{i=1}^{m} \frac{48 \binom{K'}{2}^2 \ln T}{\Delta_{\min}^i}. \qquad \blacksquare$$

The bound in Eq. (12) is tighter than the one obtained by directly applying [20].

### D.4 Comparison

**Estimator.** Let $\hat{\mu}_{i,t}$ be the estimator computed in Algo. 3 by end of round $t$ and $\hat{\mu}_{i,t}^r$ be the estimator computed with revision in Eq. (11) by end of round $t$. Both of $\hat{\mu}_{i,t}$ and $\hat{\mu}_{i,t}^r$ are unbiased estimator of $\mu_i$. However, $\hat{\mu}_{i,t}$ is a *more efficient* estimator than $\hat{\mu}_{i,t}^r$. More specifically, $\text{Var}[\hat{\mu}_{i,t}] = \mu_i(1 - \mu_i)/(\sum_{t'=1}^{t} \lfloor k_{i,t'}/2 \rfloor)$ and $\text{Var}[\hat{\mu}_{i,t}^r] = \mu_i(1 - \mu_i) \cdot (\sum_{t'=1}^{t} 1/\lfloor k_{i,t}/2 \rfloor)/(T_{i,t}^r)^2$. Here $k_{i,t}$ is the size of $\mathcal{S}_i$ in round $t$, and $T_{i,t}^r = \sum_{t'=1}^{t} \mathbb{1}\{k_{i,t'} > 1\}$. Since the harmonic mean is always not larger than arithmetic mean, i.e., $T_{i,t}^r/(\sum_{t'=1}^{t} 1/\lfloor k_{i,t'}/2 \rfloor) \leq (\sum_{t'=1}^{t} \lfloor k_{i,t'}/2 \rfloor)/T_{i,t}^r$, we conclude that $\text{Var}[\hat{\mu}_{i,t}] \leq \text{Var}[\hat{\mu}_{i,t}^r]$.

**Regret Bound.** The regret bound in Eq. (6) is tighter than the one in Eq. (12) up to $(K' - 1)/2$ factor in the $O(\ln T)$ term. The bound in Eq. (6) has a larger constant term. That's because we use a smaller confidence radius, which leads to earlier exploitation of Algo. 3 than the revised one.

### D.5 Full information feedback

In the following, we prove the constant regret bound of the Algo. 3 with feeding the empirical mean in COMMUNITYEXPLORE and making revision defined in Eq. (7).

**Proof.** We first bound $\Delta_{\boldsymbol{k}_t}$ by $\sum_{i=1}^{m} |\mu_{i,t} - \mu_i|$.

$$\Delta_{\boldsymbol{k}_t} = r_{\boldsymbol{k}^*}(\boldsymbol{\mu}) - r_{\boldsymbol{k}_t}(\boldsymbol{\mu}) = r_{\boldsymbol{k}^*}(\boldsymbol{\mu}) - r_{\boldsymbol{k}_t}(\hat{\boldsymbol{\mu}}) + r_{\boldsymbol{k}_t}(\hat{\boldsymbol{\mu}}) - r_{\boldsymbol{k}_t}(\boldsymbol{\mu})$$

$$\leq r_{\boldsymbol{k}^*}(\boldsymbol{\mu}) - r_{\boldsymbol{k}^*}(\hat{\boldsymbol{\mu}}) + r_{\boldsymbol{k}_t}(\hat{\boldsymbol{\mu}}) - r_{\boldsymbol{k}_t}(\boldsymbol{\mu}) \qquad (r_{\boldsymbol{k}^*}(\hat{\boldsymbol{\mu}}) \leq r_{\boldsymbol{k}_t}(\hat{\boldsymbol{\mu}}))$$

$$\leq |r_{\boldsymbol{k}^*}(\boldsymbol{\mu}) - r_{\boldsymbol{k}^*}(\hat{\boldsymbol{\mu}})| + |r_{\boldsymbol{k}_t}(\hat{\boldsymbol{\mu}}) - r_{\boldsymbol{k}_t}(\boldsymbol{\mu})|$$

$$\leq \sum_{i=1}^{m} \left(\binom{k_i^*}{2} + \binom{k_{i,t}}{2}\right) |\hat{\mu}_{i,t-1} - \mu_i|.$$

Leverage the fact that $\sum_{i=1}^{m} \binom{k_{i,t}}{2} \leq \binom{K'}{2}$. If $|\hat{\mu}_{i,t-1} - \mu_i| < \frac{\Delta_{\min}}{K'(K'-1)}$, then

$$\Delta_{\boldsymbol{k}_t} \leq \sum_{i=1}^{m} \left(\binom{k_i^*}{2} + \binom{k_{i,t}}{2}\right) \frac{\Delta_{\min}}{K'(K'-1)} < \Delta_{\min},$$

which means $\Delta_{\boldsymbol{k}_t} = 0$. Hence,

$$\Pr\left(\Delta_{\boldsymbol{k}_t} > 0\right) \leq \sum_{i=1}^{m} \Pr\left(|\hat{\mu}_{i,t-1} - \mu_i| \geq \frac{\Delta_{\min}}{K'(K'-1)}\right)$$

$$\leq \sum_{i=1}^{m} 2e^{-2(T_{i,t-1}/2)\Delta_{\min}^2/(K'(K'-1))^2}. \qquad \text{(Theorem 3.2 in [9])}$$

The second line of above inequality using Theorem 3.2 in [9]. Note that the $T_{i,t-1}$ member pairs using for collision counting are not independent with each other. We need to construct a *dependence graph $G$* to model their dependence. The dependence graph here is just a line with $T_{i,t-1}$ nodes. Since the fractional chromatic number of the dependence graph is 2, we have a $1/2$ factor for $T_{i,t-1}$ in the exponential. The regret is bounded as

$$\text{Reg}_{\boldsymbol{\mu}}(T) \leq \sum_{t=1}^{T}\sum_{i=1}^{m} \Delta_{\boldsymbol{k}_t} 2e^{-T_{i,t-1}\Delta_{\min}^2/(K'(K'-1))^2}$$

$$\leq 2\Delta_{\max} + \sum_{i=1}^{m}\sum_{t=3}^{T} \Delta_{\boldsymbol{k}_t} 2e^{-(t-2)\Delta_{\min}^2/(K'(K'-1))^2} \qquad (T_{i,t-1} \geq t-2)$$

$$\leq 2\Delta_{\max} + 2m\Delta_{\max}\int_{t=0}^{\infty} e^{-t\Delta_{\min}^2/(K'(K'-1))^2}\,\mathrm{d}t$$

$$\leq \left(2 + 8me^2\binom{K'}{2}^2/\Delta_{\min}^2\right)\Delta_{\max}. \qquad \blacksquare$$

## E  Regret Analysis for Adaptive Problem

### E.1  Transition probability list of policy $\pi^t$

Similar to the discussion in Section B.2.1, we define a transition probability list $\mathcal{P}(\pi^t, \psi)$ for the policy $\pi^t$ and write the reward function $r_{\pi^t}(\boldsymbol{\mu})$ with $\mathcal{P}(\pi^t, \emptyset)$.

**Definition.** Assume the initial partial realization is $\psi$. Let $\boldsymbol{s}_0$ be the status corresponding to $\psi$. Recall that $\boldsymbol{s}_0 = (s_{1,0}, \ldots, s_{m,0}) = (1 - \mu_1 c_1(\psi), \ldots, 1 - \mu_m c_m(\psi))$. At the first step, policy $\pi^t$ chooses community $i_0^* = \arg\max_{i\in[m]} 1 - c_i(\psi)\underline{\mu}_{i,t}$. With probability $q_0^{\pi^t} := c_{i_0^*}(\psi)\mu_{i_0^*}$, the communities stay at the same status $\boldsymbol{s}_0$. With probability $p_0^{\pi^t} := 1 - c_{i_0^*}(\psi)\mu_{i_0^*}$, the communities transit to next status $\boldsymbol{s}_1 := \boldsymbol{s}_0 - \mu_{i_0^*}\boldsymbol{I}$. Note that

$$1 - c_i(\psi)\underline{\mu}_{i,t} = \frac{\mu_i - (1-s_{i,0})\underline{\mu}_{i,t}}{\mu_i} = \frac{\underline{\mu}_{i,t}}{\mu_i}s_{i,0} + \frac{\mu_i - \underline{\mu}_{i,t}}{\mu_i}.$$

We recursively define $\boldsymbol{s}_{k+1}$ as $\boldsymbol{s}_k - \mu_{i_k^*}\boldsymbol{I}_{i_k^*}$ where $i_k^* \in \max_{i\in[m]} (\underline{\mu}_{i,t}/\mu_i)s_{i,k} + (\mu_i - \underline{\mu}_{i,t})/\mu_i$. The transition probability $p_k^{\pi^t} := s_{i_k^*,k}$. We define the transition probability list $\mathcal{P}(\pi^t, \psi) = (p_0^{\pi^t}, \ldots, p_D^{\pi^t})$ where $D = \sum_{i=1}^{m}(d_i - c_i(\psi))$ is the number of distinct member we haven't meet under the partial realization $\psi$. Note that it is possible that $p_k^{\pi^t} = 0$. In this case, there is already no unmet members in $i_k^*$. The communities will be stuck in status $\boldsymbol{s}_k$ since the policy $\pi^t$ always chooses community $i_k^*$ to explore after the communities reach status $\boldsymbol{s}_k$. Hence, if $k$ is the smallest index such that $p_k^{\pi^t} = 0$, we define $p_{k'}^{\pi^t} = 0$ for all $k' > k$.

**Compute** $\mathcal{P}(\pi^t, \psi)$. Define $\mathcal{B}_i(\psi) = \{1 - c_i(\psi)\mu_i, 1 - (1 + c_i(\psi))\mu_i, \ldots, \mu_i, 0\}$ for $i \in [m]$. Let $b_i \in \mathcal{B}_i(\psi), b_j \in \mathcal{B}_j(\psi), i,j \in [m]$. We define a *sorting comparator* as follows.

$$\text{less}(b_i, b_j) = \mathbb{1}\{(\underline{\mu}_{i,t}/\mu_i)\cdot b_i + (\mu_i - \underline{\mu}_{i,t})/\mu_i < (\underline{\mu}_{j,t}/\mu_j)\cdot b_j + (\mu_j - \underline{\mu}_{j,t})/\mu_j\}$$

If $b_i \geq b_j$ and $\text{less}(b_i, b_j) = 1$, we can infer that $\underline{\mu}_{i,t}/\mu_i \geq \underline{\mu}_{j,t}/\mu_j$, which means the size of community $j$ is more overestimated than the size of community $i$. The overestimation leads to wrong order between $b_i$ and $b_j$ when using the comparator less. The list $\mathcal{P}(\pi^t, \psi)$ can be computed as follows. Firstly, we sort elements in $\cup_{i\in[m]}\mathcal{B}_i$ with the comparator less. Secondly, we truncate the

sorted list at the first zero elements. Thirdly, we paddle zeros at the end of list until the length is $D + 1$. All the arguments in Section B.2-B.1 about $\mathcal{P}(\pi^g, \psi)$ can be easily extended to $\mathcal{P}(\pi^t, \psi)$.

**Expected reward.** In the following, we still use the extended definition of reward

$$R(\boldsymbol{k}, \phi) = f\left(\sum_{i=1}^{m} \left| \bigcup_{\tau=1}^{k_i} \{\phi(i, \tau)\} \right| \right),$$

where $f$ is a non-decreasing function. We can write the reward function $r_{\pi^t}(\boldsymbol{\mu})$ as

$$r_{\pi^t}(\boldsymbol{\mu}) = \sum_{j=0}^{\min\{K, \sum_{i=1}^{m} d_i\}} f(j) \times p_0^{\pi^t} \times \cdots \times p_{j-1}^{\pi^t} \times L(\{q_0^{\pi^t}, \dots, q_j^{\pi^t}\}, K - j).$$

Here $p_j^{\pi^t}$ is element in $\mathcal{P}(\pi^t, \emptyset)$, $q_j^{\pi^t} := 1 - p_j^{\pi^t}$, and $K$ is the budget.

### E.2 Proof framework

**Notations.** Let $D = \sum_{i=1}^{m} d_i$ in this part. Let $\mathcal{P}(\pi^g, \emptyset) = (p_0^{\pi^g}, \dots, p_D^{\pi^g})$ and $\mathcal{P}(\pi^t, \emptyset) = (p_0^{\pi^t}, \dots, p_D^{\pi^t})$. According to Corollary 1, we know that $\mathcal{P}(\pi^g, \emptyset)$ can be obtained by sorting $\cup_{i \in [m]}\{1, 1 - \mu_i, 1 - 2\mu_i, \dots, \mu_i\} \cup \{0\}$. Here we define another list $\tilde{\mathcal{P}}(\pi^g)$ which is obtained by sorting $\cup_{i \in [m]}\{(i, 1), (i, 1 - \mu_i), \dots, (i, \mu_i)\}$ via comparing the second value in the pair. Let $U_{i,k}$ denote how many times pair $(i, \cdot)$ appears in the first $k$ positions in the list $\tilde{\mathcal{P}}(\pi^g)$. The value $U_{i,k}$ satisfies that $p_k^{\pi^g} = \max_{i=1}^{m} 1 - U_{i,k}\mu_i$. Note that the definition of $U_{i,k}$ are equivalent to the one defined in the main text.

**Theorem 5.** *Algo. 3 with adaptive exploration method has regret as follows.*

$$Reg_{\boldsymbol{\mu}}(T) \le \left( \sum_{i=1}^{m} \sum_{k=m+1}^{\min\{K,D\}} \frac{6\Delta_{\max}^{(k)}}{(\Delta_{\min}^{i,k})^2} \right) \ln T + \frac{\lfloor \frac{K'}{2} \rfloor \pi^2}{3} \sum_{i=1}^{m} \sum_{k=m+1}^{\min\{K,D\}} \Delta_{\max}^{(k)}. \tag{8}$$

**Proof.** When $\boldsymbol{\mu}_t$ is close to $\boldsymbol{\mu}$, the list $\mathcal{P}(\pi^t, \emptyset)$ is similar to the list $\mathcal{P}(\pi^g, \emptyset)$, which indicates the reward gap $r_{\pi^g}(\boldsymbol{\mu}) - r_{\pi^t}(\boldsymbol{\mu})$ is small. Let $\mathbb{1}_{i,k}(\boldsymbol{\mu}_t)$ be the indicator that takes value 1 when $\mathcal{P}(\pi^g, \emptyset)$ and $\mathcal{P}(\pi^t, \emptyset)$ are the same for the first $k$ elements, and different at the $(k+1)$-th elements (i.e., $p_j^{\pi^g} = p_j^{\pi^t}$ for $0 \le j \le k - 1$ and $p_k^{\pi^g} \ne p_k^{\pi^t}$) with condition $p_k^{\pi^t} = 1 - U_{i,k}\mu_i$. Note that the first $m$ elements in $\mathcal{P}(\pi^t, \emptyset)$ and $\mathcal{P}(\pi^g, \emptyset)$ equal to 1. Then the reward gap at round $t$ is

$$\Delta_{\pi^t} = r_{\pi^g}(\boldsymbol{\mu}) - r_{\pi^t}(\boldsymbol{\mu}) = \sum_{i=1}^{m} \sum_{k=m+1}^{\min\{K,D\}} \mathbb{1}_{i,k}(\boldsymbol{\mu}_t) \cdot \Delta_{\max}^{i,k},$$

where $\Delta_{\max}^{i,k}$ is the maximum reward gap among all possible $\boldsymbol{\mu}_t$ such that $\mathbb{1}_{i,k}(\underline{\boldsymbol{\mu}}_t) = 1$, i.e.,

$$\Delta_{\max}^{i,k} = \max_{\forall \boldsymbol{\mu}_t, \mathbb{1}_{i,k}(\boldsymbol{\mu}_t)=1} r_{\pi^g}(\boldsymbol{\mu}) - r_{\pi^t}(\boldsymbol{\mu}).$$

Note that

$$\Delta_{\max}^{i,k} \le \sum_{j=k}^{\min\{K,D\}} f(j) \times p_0^{\pi^g} \times \cdots \times p_{j-1}^{\pi^g} \times L(\{1 - p_0^{\pi^g}, \cdots, 1 - p_j^{\pi^g}\}, K - j).$$

The expected cumulative regret can be expanded as

$$\text{Reg}_{\boldsymbol{\mu}}(T) = \mathbb{E}_{\Phi_1, \cdots, \Phi_T}\left[ \sum_{t=1}^{T} \Delta_{\pi^t} \right] \le \sum_{t=1}^{T} \mathbb{E}_{\Phi_1, \cdots, \Phi_{t-1}}\left[ \sum_{k=m+1}^{\min\{K,D\}} \sum_{i=1}^{m} \mathbb{1}_{i,k}(\boldsymbol{\mu}_t) \times \Delta_{\max}^{i,k} \right]$$

$$= \sum_{i=1}^{m} \sum_{k=m+1}^{M} \Delta_{\min}^{i,k} \mathbb{E}_{\Phi_1, \cdots, \Phi_{t-1}}\left[ \sum_{t=1}^{T} \mathbb{1}_{i,k}(\boldsymbol{\mu}_t) \right].$$

Our next step is bound $\mathbb{E}_{\Phi_1,\cdots,\Phi_{t-1}}\left[\sum_{t=1}^{T}\mathbb{1}_{i,k}(\boldsymbol{\mu}_t)\right]$. We rewrite the indicator $\mathbb{1}_{i,k}(\boldsymbol{\mu}_t)$ as:

$$\mathbb{1}_{i,k}(\boldsymbol{\mu}_t)=\mathbb{1}_{i,k}(\boldsymbol{\mu}_t)\mathbb{1}\{T_{i,t-1}\leq l_{i,k}\}+\mathbb{1}_{i,k}(\boldsymbol{\mu}_t)\mathbb{1}\{T_{i,t-1}>l_{i,k}\},$$

where $l_{i,k}$ is a problem-specific constant. In Lemma 11, we show that the probability we choose a wrong community when community $i$ is probed enough times (i.e., $T_{i,t-1}>l_{i,k}$) is very small. Based on the lemma, the regret corresponding to the event $\mathbb{1}\{T_{i,t-1}>l_{i,k}\}$ is bounded as follows.

$$\sum_{i=1}^{m}\sum_{k=m+1}^{\min\{K,D\}}\Delta_{\min}^{i,k}\mathbb{E}_{\Phi_1,\cdots,\Phi_T}\left[\sum_{t=1}^{T}\mathbb{1}_{i,k}(\boldsymbol{\mu}_t)\mathbb{1}\left\{T_{i,t-1}>l_{i,k}\right\}\right]\leq\frac{\lfloor\frac{K'}{2}\rfloor\pi^2}{3}\sum_{i=1}^{m}\sum_{k=m+1}^{\min\{K,D\}}\Delta_{\max}^{i,k}.$$

On the other hand, the regret associated with the event $\mathbb{1}\{T_{i,t-1}\leq l_{i,k}\}$ is trivially bounded by $\sum_{i=1}^{m}\sum_{k=m+1}^{K}\Delta_{\max}^{i,k}l_{i,k}$. In conclusion, the expected cumulative regret is bound as

$$\mathrm{Reg}_{\boldsymbol{\mu}}(T)\leq\sum_{i=1}^{m}\sum_{k=m+1}^{K}\Delta_{\max}^{i,k}\mathbb{E}_{\Phi_1,\cdots,\Phi_T}\left[\sum_{t=1}^{T}\mathbb{1}_{k,t}(\boldsymbol{\mu}_t)\right]$$

$$\leq\sum_{i=1}^{m}\sum_{k=m+1}^{K}\Delta_{\max}^{i,k}l_{i,k}+\frac{\lfloor\frac{K'}{2}\rfloor\pi^2}{3}\sum_{i=1}^{m}\sum_{k=m+1}^{\min\{K,D\}}\Delta_{\max}^{i,k}$$

$$\leq\left(\sum_{i=1}^{m}\sum_{k=m+1}^{K}\frac{6\Delta_{\max}^{i,k}}{(\Delta_{\min}^{i,k})^2}\right)\ln T+\frac{\lfloor\frac{K'}{2}\rfloor\pi^2}{3}\sum_{i=1}^{m}\sum_{k=m+1}^{\min\{K,D\}}\Delta_{\max}^{i,k}.$$

Note $\Delta_{\max}^{(k)}\geq\max_{i\in[m]}\Delta_{\max}^{i,k}$. This completes the proof. ∎

**Lemma 11.** *For all $k\leq\{M,\sum_{i=1}^{m}d_i\}$, we have*

$$\mathbb{E}_{\Phi_1,\ldots,\Phi_T}\left[\sum_{t=1}^{T}\mathbb{1}_{i,k}(\boldsymbol{\mu}_t)\mathbb{1}\{T_{i,t-1}>l_{i,k}\}\right]\leq\frac{\lfloor\frac{K'}{2}\rfloor\pi^2}{3}, \tag{13}$$

*where $l_{i,k}$ is defined as $l_{i,k}:=6\ln T/(\Delta_{\min}^{i,k})^2$.*

**Proof.** The following proof is similar to the that for the traditional Upper Confidence Bound (UCB) algorithm [1]. In the following, we define $i_k^*=\max_{i\in[m]}1-U_{i,k}\mu_i$.

$$\sum_{t=1}^{T}\mathbb{1}_{i,k}(\boldsymbol{\mu}_t)\mathbb{1}\{T_{i,t-1}>l_{i,k}\}=\sum_{t=l_{i,k}+1}^{T}\mathbb{1}_{i,k}(\boldsymbol{\mu}_t)\mathbb{1}\{T_{i,t-1}>l_{i,k}\}$$

$$\leq\sum_{t=l_{i,k}+1}^{T}\mathbb{1}\{(\hat{\mu}_{i,t-1}-\rho_{i,t-1})U_{i,k})<(\hat{\mu}_{i_k^*,t-1}-\rho_{i_k^*,t-1})U_{i_k^*,k},T_{i,t-1}>l_{i,k}\}.$$

When $T_{i_k,t-1}>l_{i,k}\triangleq\frac{6\ln T}{(\Delta_{\min}^{i,k})^2}$, we have

$$\rho_{i,t-1}=\sqrt{\frac{3\ln t}{2T_{i,t-1}}}<\frac{\Delta_{\min}^{i,k}}{2}\Rightarrow\underbrace{\mu_{i_k^*}U_{i_k^*,k}<(\mu_i-2\rho_{i,t-1})U_{i,k}}_{i\text{ and }i_k^*\text{ are distinguishable with high prob.}}.$$

If $i\neq i_k^*$ exists such that

$$\hat{\mu}_{i_k^*,t-1}-\rho_{i_k^*,t-1}<\mu_{i_k^*},\text{ and }\hat{\mu}_{i,t-1}+\rho_{i,t-1}>\mu_i,$$

we have

$$(\hat{\mu}_{i_k^*,t-1}-\rho_{i_k^*,t-1})U_{i_k^*,k}<\mu_{i_k^*}U_{i_k^*,k}<(\mu_i-2\rho_{i,t-1})U_{i,k}<(\hat{\mu}_{i,t-1}-\rho_{i,t-1})U_{i,k},$$

which contradicts with $(\hat{\mu}_{i,t-1} - \rho_{i,t-1})U_{i,k} < (\hat{\mu}_{i_k^*,t-1} - \rho_{i_k^*,t-1})U_{i_k^*,k}$. Hence when $T_{i,t-1} > l_{i,k}$, we have

$$\left\{ (\hat{\mu}_{i,t-1} - \rho_{i,t-1})U_{i,k} < (\hat{\mu}_{i_k^*,t-1} - \rho_{i_k^*,t-1})U_{i_k^*,k} \right\}$$
$$\subseteq \left\{ \hat{\mu}_{i,t-1} + \rho_{i,t-1} \le \mu_i \text{ or } \hat{\mu}_{i_k^*,t-1} - \rho_{i_k^*,t-1} \ge \mu_{i_k^*} \right\}$$

Using the union bound, we have

$$\Pr\left( (\hat{\mu}_{i,t-1} - \rho_{i,t-1})U_{i,k} < (\hat{\mu}_{i_k^*,t-1} - \rho_{i_k^*,t-1})U_{i_k^*,k} \right)$$
$$\le \Pr\left( \hat{\mu}_{i,t-1} + \rho_{i,t-1} \le \mu_i \text{ or } \hat{\mu}_{i_k^*,t-1} - \rho_{i_k^*,t-1} \ge \mu_{i_k^*} \right)$$
$$\le \Pr\left( \hat{\mu}_{i,t-1} + \rho_{i,t-1} \le \mu_i \right) + \Pr\left( \hat{\mu}_{i_k^*,t-1} - \rho_{i_k^*,t-1} \ge \mu_{i_k^*} \right).$$

Therefore, we can conclude that

$$\mathbb{E}_{\Phi_1,\dots,\Phi_T}\left[ \sum_{t=1}^{T} \mathbb{1}_{i,k}(\underline{\mu}_t)\mathbb{1}\{T_{i,t-1} > l_{i,k}\} \right]$$
$$\le \sum_{t=l_{i,k}+1}^{T} \mathbb{1}\{ (\hat{\mu}_{i,t-1} - \rho_{i,t-1})U_{i,k} < (\hat{\mu}_{i_k^*,t-1} - \rho_{i_k^*,t-1})U_{i_k^*,k}, T_{i,t-1} > l_{i,k} \}.$$
$$\le \sum_{t=l_{i,k}+1}^{T} \Pr\{ \hat{\mu}_{i,t-1} + \rho_{i,t-1} \le \mu_i \} + \Pr\{ \hat{\mu}_{i_k^*,t-1} - \rho_{i_k^*,t-1} \ge \mu_{i_k^*} \}$$
$$\le \sum_{t=l_{i,k}+1}^{T} \left( \sum_{T_{i,t-1}=l_{i,k}+1}^{t\lfloor\frac{K'}{2}\rfloor} \Pr\{ \hat{\mu}_{i,t-1} + \rho_{i,t-1} \le \mu_i | T_{i,t-1} \} \right.$$
$$\left. + \sum_{T_{i_k^*,t-1}=1}^{t\lfloor\frac{K'}{2}\rfloor} \Pr\left( \hat{\mu}_{i_k^*,t-1} - \rho_{i_k^*,t-1} \ge \mu_{i_k^*} | T_{i_k^*,t-1} \right) \right)$$
$$\le \sum_{t=1}^{\infty} 2t\left\lfloor \frac{K'}{2} \right\rfloor \times t^{-3} = 2\left\lfloor \frac{K'}{2} \right\rfloor \sum_{t=1}^{\infty} t^{-2} = \frac{\left\lfloor \frac{K'}{2} \right\rfloor \pi^2}{3}. \qquad \blacksquare$$

### E.3  Full information feedback

If we feed the empirical mean in the exploration oracle, then the policy $\pi^t$ is determined by $\hat{\mu}_t$. Similarly, we can define the event $\mathbb{1}_{i,k}(\hat{\mu}_t)$ by replacing $\underline{\mu}_t$ with $\hat{\mu}$ in Section E.1-E.2.

**Lemma 12.** *If we make revisions defined in Eq. (7) to Algo. 3 and feed the empirical mean in* COMMUNITYEXPLORE *to explore communities adaptively, then for all community $C_i$ and $k \le \{K, \sum_{i=1}^{m} d_i\}$, we have*

$$\mathbb{E}_{\Phi_1,\dots,\Phi_T}\left[ \sum_{t=2}^{T} \mathbb{1}_{i,k}(\hat{\mu}_t) \right] \le \frac{2}{\varepsilon_{i,k}^4}, \tag{14}$$

*where $\varepsilon_{i,k}$ is defined as (here $i_k^* \in \arg\min_{i\in[m]} \mu_i U_{i,k}$)*

$$\varepsilon_{i,k} \triangleq \frac{\mu_i U_{i,k} - \mu_{i_k^*} U_{i_k^*,k}}{U_{i,k} + U_{i_k^*,k}} \text{ for } i \ne i_k^* \text{ and } \varepsilon_{i,k} = \infty \text{ for } i = i_k^*.$$

*Proof.* We first bound the probability of the following event by relating $\mathbb{1}_{i,k}(\hat{\mu}_t)$ with the event that both $\mu_{i,t-1}$ and $\mu_{i_k,t-1}$ in the confidence interval $\varepsilon_{i,k}$.

$$\mathbb{1}_{i,k}(\hat{\mu}_t) \le \mathbb{1}\left\{ \hat{\mu}_{i,t-1}U_{i,k} < \hat{\mu}_{i_k^*,t-1}U_{i_k^*,k} \right\}.$$

If $i \ne i_k^*$ such that

$$\hat{\mu}_{i,t-1} > \mu_i - \varepsilon_{i,k}, \text{ and } \hat{\mu}_{i_k^*,t-1} < \mu_{i_k^*} + \varepsilon_{i,k},$$

then

$$\hat{\mu}_{i,t-1}U_{i,k} > (\mu_i - \varepsilon_{i,k})U_{i,k} = (\mu_{i_k^*} + \varepsilon_{i,k})U_{i_k^*,k} > \hat{\mu}_{i_k^*,t-1}U_{i_k^*,k},$$

which contradicts with that $\hat{\mu}_{i,t-1}U_{i,k} < \hat{\mu}_{i_k^*,t-1}U_{i_k^*}$. Here $(\mu_i - \varepsilon_{i,k})U_{i,k} = (\mu_{i_k^*} + \varepsilon_{i^*,k})U_{i_k^*,k}$ can be derived from the definition of $\varepsilon_{i,k}$. Therefore

$$\mathbb{1}\left\{\hat{\mu}_{i,t-1}U_{i,k} < \hat{\mu}_{i_k^*,t-1}U_{i_k^*,k}\right\}$$
$$\leq \mathbb{1}\left\{\hat{\mu}_{i,t-1} \leq \mu_i - \varepsilon_{i,k} \text{ or } \hat{\mu}_{i_k^*,t-1} \geq \mu_{i_k^*} + \varepsilon_{i,k}\right\}.$$

With above equation and the concentration bound in [9], the expectation $\mathbb{E}_{\Phi_1,\ldots,\Phi_T}\left[\sum_{t=2}^T \mathbb{1}_{i,k}(\hat{\boldsymbol{\mu}}_t)\right]$ can be bounded as

$$\mathbb{E}_{\Phi_1,\ldots,\Phi_T}\left[\sum_{t=2}^T \mathbb{1}_{i,k}(\hat{\boldsymbol{\mu}}_t)\right]$$

$$\leq \sum_{t=2}^T \Pr\left\{\hat{\mu}_{i,t-1}U_{i,k} < \hat{\mu}_{i_k^*,t-1}U_{i_k^*,k}\right\}$$

$$\leq \sum_{t=2}^T \Pr\left\{\hat{\mu}_{i,t-1} \leq \mu_i - \varepsilon_{i,k}\right\} + \Pr\left\{\hat{\mu}_{i_k^*,t-1} \geq \mu_{i_k^*} + \varepsilon_{i,k}\right\}$$

$$\leq \sum_{t=2}^T \left(\sum_{T_{i,t-1}=t-1}^{t\lfloor K'/2 \rfloor} e^{-\varepsilon_{i,k}^2 T_{i,t-1}} + \sum_{T_{i_k^*,t-1}=t-1}^{t\lfloor K'/2 \rfloor} e^{-\varepsilon_{i,k}^2 T_{i_k^*,t-1}}\right)$$

$$\leq 2\sum_{t=1}^T \sum_{s=t}^\infty e^{-s\varepsilon_{i,k}^2} \leq 2\sum_{t=1}^T \frac{e^{-t\varepsilon_{i,k}^2}}{\varepsilon_{i,k}^2} \leq \frac{2}{\varepsilon_{i,k}^4}. \qquad \blacksquare$$

## F   Experimental Evaluation

In this section, we conduct simulations to validate the theoretical results claimed in the main text and provide some insight for future research.

### F.1   Offline Problems

In this part, we show some simulation results for the offline problems.

**Performance of Algorithm 1**. In Fig. 3, we show that the allocation lower bound $\boldsymbol{k}^-$ and upper bound $\boldsymbol{k}^+$ are close to the optimal budget allocation. From Fig. 3, we observe that the $L1$ distance between $\boldsymbol{k}^*$ and $\boldsymbol{k}^-$ (or $\boldsymbol{k}^+$) is around $m/2$, which means the average time complexity of Algorithm 1 is $\Theta((m\log m)/2)$.

**Reward v.s. Budget**. We show the relationship between the reward (i.e., the number of distinct members) and the given budget in Fig. 4. From Fig. 4, we can draw the following conclusions.

- The performance of the four methods are ranked as: "Adaptive Opt.", *"Non-adaptive Opt."*, *"Proportional to Size"*, *"Random Allocation"*. This validate our optimality results in Sec. 3.

- The method "*Proportional to Size*" and "*Non-adaptive Opt.*" have similar performance. It is an intuitive idea to allocate budgets proportional to the community sizes. The simulation results also demonstrate the efficiency of such budget allocation method. In the following, we analyze the reason theoretically. Recall the definition of $\boldsymbol{k}^-$ as follows.

$$k_i^- = \frac{(K-m)/\ln(1-\mu_i)}{\sum_{j=1}^m 1/\ln(1-\mu_j)}.$$

When $\mu_i \ll 1$, we have $\ln(1-\mu_i) \approx -\mu_i$. Hence,

$$k_i^- \approx \frac{(K-m)d_i}{\sum_{j=1}^m d_j}.$$

Besides, the L1 distance between $\boldsymbol{k}^*$ and $\boldsymbol{k}^-$ is smaller than $m$. We can conclude that the budget allocation proportional to size is close to the optimal budget allocation. Fig. 6 also validates this conclusion.

Figure 3: The $L1$ distance between $\boldsymbol{k}^*$ and $\boldsymbol{k}^-$, $\boldsymbol{k}^+$ under different community size distributions. Here $\mathcal{U}\{2, 26\}$ is the discrete uniform distribution between 2 and 26. $\mathcal{G}(0.1)$ is the geometric distribution with success probability $0.1$ on the support set $\{2, 3, \dots\}$. $\Gamma(\alpha, \beta)$ is the gamma distribution with shape $\alpha$ and rate $\beta$. We discretize the support set of the gamma distribution and add 2 to all the values in the support set to ensure that the minimum size of communities is 2. The budget $K$ is a random number between $m + 1$ and $\sum_i d_i$. We run the simulations for 1000 times for each data point.

- The reward gap between "*Non-adaptive Opt.*" and "*Adaptive Opt.*" increases first and then decreases, as shown in Fig. 5.

**Budget Allocation Comparison**. Fig. 6 and Fig 5 show the budget allocation of non-adaptive optimal method and adaptive optimal method. Fig. 5 shows that the adaptive optimal method use the budget more efficiently.

## F.2 Online Problems

In the following, we show the simulation results for the online, non-adaptive problem. The simulation results for online, adaptive are similar. Hence, we only present the results for online, non-adaptive problems. Fig. 7 shows the regret of three different learning methods. For illustration purpose, we set the community sizes as
$bmd = (2, 3, 5, 6, 8, 10)$. From Fig. 7, we can draw the following conclusions.

- If we feed the empirical mean into the oracle, the regret grows linearly.
- The regret of CLCB algorithm is bounded logarithmically, as proved in Thm. 3.
- The regret under full information feedback setting is bounded as a problem related constant, as proved in Thm. 4.

(a) Community size distributions

Random Allocation — Proportional to Size — Non-adaptive Opt. — Adaptive Opt.

(b) Reward of different methods

Figure 4: Reward v.s. Budget. In the first row, we show three different size distributions of $m = 100$ communities. In the second row, we show the reward of four different budget allocation methods. Here *"Random Allocation"* represents random budget allocation (sum up to $K$). *"Proportional to Size"* method allocates budget proportional to the community sizes. *"Non-adaptive Opt."* corresponds to the optimal budget allocation obtained by the greedy method. *"Adaptive Opt."* means we explore the communities with greedy adaptive policy $\pi^g$. The simulations are run for 200 times for each data point on the budget-reward curve.

Figure 5: Actually used budget. we only show the results for the community size configuration generated by $\mathcal{G}(0.1)$, as shown in the first row of Fig. 4. The legend labels have the same meaning as in Fig. 6

Figure 6: Comparison of different budget allocation methods. The distribution of community sizes generated by the geometric distribution with success probability $0.1$, as shown in the first row of Fig. 4. The legend label "$\boldsymbol{k}^*$" represents the optimal budget allocation. The "truncated $\pi^g$" means we stop the greedy adaptive process if all the members are found. The "truncated $\boldsymbol{k}^*$" means we stop the non-adaptive exploration of community $C_i$ if all the members of $C_i$ are found. Each data point is an average of 1000 simulations.

Figure 7: Comparison of different learning algorithms. The sizes of communities are $\boldsymbol{d} = (2, 3, 5, 6, 8, 10)$. Here the label *Empirical mean* represents feeding the empirical mean into the oracle directly. The regret/error line plots are average of 100 simulations.