[Reviews · NeurIPS 2018]

Reviewer 1



This paper proposes a variant of community exploration problem in which the explorer allocates limited budget to explore communities so as to maximize the number of members he/she could meet. The authors provide a complete and systematic study of different settings (offline optimization vs. online learning, non-adaptive learning vs. adaptive learning) of the community exploration problem. The authors prove that, for offline learning, greedy policy obtains an optimal budget allocation in both non-adaptive and adaptive setting. For online learning, the key is to estimate the community sizes. With multi-round explorations, the authors propose a combinatorial lower confidence bound (CLCB) algorithm that achieves the logarithmic regret bounds. Furthermore, by combining the feedback from different rounds (full information feedback), the algorithm can achieve a constant regret bound. Pros: This paper is clear and easy to follow. I spent much time going through all proofs in appendix and don't see any major errors. One contribution of this paper is that the authors propose and formulate the community exploration problem in rigorous mathematical language. As for algorithmic part, I especially like how systematic the paper solves community exploration problem. The authors discuss various scenarios and give full and complete analysis. The conclusions are insightful. Usually for adaptive submodular problem, a greedy policy can only achieve (1-1/epsilon) approximation. Here the analysis show greedy policy is guaranteed optimal for this type of problems. Cons: A minor drawback of this paper is that it lacks experimental verification. For a simulated setting, I would not be surprised if the experiments match exactly the theoretical bounds. It may be more of interest to see a real-world application. However, given the paper's solid theoretical contributions, missing experiments is acceptable. I believe this paper qualifies for NIPS requirement.

Reviewer 2



The paper introduces the community exploration problem, which can be seen as a resource allocation problem whose goal is to divide resources among several disjoint sub-communities such that the number of distinct community members encountered is maximized. The paper contributes with algorithms (and an extensive theoretical analysis thereof) for the offline setting of the community exploration problem when the sizes of communities are known, as well as for an online setting, when the sizes of the communities need to be learned. The paper is well written, I enjoyed reading it. The problem is interesting and the results are significant and sound (as far as I could check). Just one comment: there are no empirical tests to allow to observe the behavior of the algorithm on a real-world data problem. For instance, in what application are communities really disjoint? Even for a more simulated setting, what is the difference in empirical performance between adaptive and non-adaptive strategies? Despite the lack of experiments, I vote for accepting the submission for the nice, clear explanations (of the problem and of the proposed algorithms) and for the in-depth theoretical study.

Reviewer 3



Summary In the submission, authors explore a new “community exploration problem”, both in an offline and online setting: An agent choose at each round t \in [K] one community among C_1,…,C_m. Then, a member is uniformly sampled (with replacement) from the chosen community. The goal for the agent is to maximize the overall number of distinct members sampled. In the offline setting, the agent knows each community size. If the allocation strategy k_1+...+k_m=K has to be given before the beginning of the game (scenario 1), then a greedy non-adaptive strategy is shown to be optimal. If the agent can adapt its allocation on the fly (scenario 2), then a greedy adaptive strategy is shown to be optimal. In the online setting, community sizes are unknown. Perhaps because tracking overall sets S_1,…,S_m of met members is costly (space & time complexity), authors consider the restricted case where only few parameters can be maintained across rounds. More precisely, they redefined the notion of round: a round is composed of K sub-rounds in which the agent performs a strategy k_1+...+k_m=K (restricted to scenario 1 or 2, depending on the setting), and updates parameters at the end of these K sub-rounds, to then go to the next round, with in memory only these parameters. Authors provide CUCB-based algorithms, that call offline greedy algorithms, and prove a O(log(T)) bound on regret in the setting described above. Quality The submission appears to be technically correct. The offline part is interesting as an operational research problem, and proofs are well-written. I think this part represent the major contribution of the paper. Clarity For the online part, I regret that the setting was not clearly explained. We do not know what is allowed for the learning algorithm. I think this is an important step (particularly if one wants to compete with provided solutions) that has been largely neglected. Even if the setting can be considered as “implicitly defined” from the regret definition, no justification are given for the (unnatural) restriction considered (maybe the “complexity” argument I gave above could be a reason, but I don’t find it realistic). The more natural online setting where collision with previous rounds can be used is only considered in the future work section. It would have been better to see it before, in order to contrast with the restriction considered. Originality There is a significant lack of related work. For instance, the setting described in [1] is clearly related and should appear in the paper. The online setting analysis is pretty classic. Significance Offline results can be considered as important, the optimality of greedy algorithm being not trivial. Provided application are also relevant. I am not convinced for the online part. I don’t see a clear and well justified setting definition that support the notion of regret used in the paper. [1] S. Bubeck, D. Ernst, and A. Garivier. Optimal discovery with probabilistic expert advice: finite time analysis and macroscopic optimality. Journal of Machine Learning Research, 14:601–623, 2013.